# CD169+ macrophages orchestrate plasmacytoid dendritic cell arrest and retention for optimal priming in the bone marrow of malaria-infected mice

**Jamie Moore-Fried[1†], Mahinder Paul[1†], Zhixin Jing[1,2], David Fooksman[1,2]\*, Gregoire Lauvau[1]\***

[1]Albert Einstein College of Medicine, Department of Microbiology and Immunology, Bronx, United States; [2]Albert Einstein College of Medicine, Department of Pathology, New York, United States

**\*For correspondence:**
david.fooksman@einsteinmed.edu (DF);
gregoire.lauvau@einsteinmed.edu (GL)

[†]These authors contributed equally to this work

**Competing interest:** The authors declare that no competing interests exist.

**Abstract** Plasmacytoid dendritic cells (pDCs) are the most potent producer of type I interferon (IFN), but how pDC is primed in vivo is poorly defined. Using a mouse model of severe malaria, we have previously established that upon priming by CD169+ macrophages (MPs), pDC initiates type I IFN-I secretion in the bone marrow (BM) of infected mice via cell-intrinsic TLR7 sensing and cell-extrinsic STING sensing. Herein we show that CD169+ MP and TLR7 sensing are both required for pDC arrest during priming, suggesting CD169+ MP are the source of TLR7 ligands. We establish that TLR7 sensing in pDC and chemotaxis are both required for pDC arrest and functional communication with CD169+ MP in the BM. Lastly, we demonstrate that STING sensing in CD169+ MP control pDC initiation of type I IFN production while also regulating pDC clustering and retention/egress from the BM. Collectively, these results link pDC acquisition of type I IFN-secreting capacity with changes in their motility, homing and interactions with CD169+ MP during infection. Thus, targeting this cellular interaction may help modulate type I IFN to improve outcomes of microbial infections and autoimmune diseases.

## Editor's evaluation

This study presents valuable mechanistic findings demonstrating the requirement of CD169+ macrophage intrinsic STING signaling in regulating plasmacytoid DC motility/arrest and their activation. The evidence supporting the claims of the authors is solid, and their data suggest that adequate type I IFN production by pDC may also require macrophage intrinsic STING signaling. The work will be of great interest to the immunology community.

## Introduction

Plasmacytoid dendritic cells (pDCs) (*Colonna et al., 2004*; *Liu, 2005*; *Reizis, 2019*) are equipped with the unique ability to rapidly secrete up to 1000 times more type I interferon (IFN) – both IFNα and IFNβ – than any other cell (*Cella et al., 1999*; *Siegal et al., 1999*; *Asselin-Paturel et al., 2001*). This is accounted for by several pDC-intrinsic and -extrinsic mechanisms. pDC-intrinsic features include the constitutive expression of the endosomal Toll-like receptors (TLR-) 7 and 9, which, respectively, recognize single stranded RNA and unmethylated CpG motifs containing DNA, leading to type I IFN production via MyD88 signaling and IRF7-mediated transcription, the constitutively highly expressed master regulator of their type I IFN producing capacity (*Honda et al., 2005b*; *Honda et al., 2005a*).

Several pDC-extrinsic mechanisms are also involved in pDC activation, during which pathogen-derived material is provided to pDC by other cells, allowing further checkpoints of pDC activation (*Reizis, 2019*). However, the precise cell–cell interactions and dynamic requirements are still not well understood.

Type I IFN is a potent immunoregulatory cytokine that modulates hundreds of genes (*Stetson and Medzhitov, 2006*; *Ivashkiv and Donlin, 2014*; *Barrat et al., 2019*). It promotes early innate immune responses by favoring cell-autonomous defense mechanisms that rapidly restrict viral escape and growth. Type I IFN signaling induces the production of proinflammatory cytokines that enable the mobilization of innate immune defenses and set the stage for adaptive immune responses by enhancing antigen processing, presenting, and costimulatory capacities of antigen presenting cells. It also directly signals to activated T and B cells, promoting expression of anti-apoptotic molecules and orchestrating their differentiation into robust effector cells. In contrast to these beneficial effects, excess or dysregulated type I IFN production that occurs in the context of auto-immune diseases, suggests that type I IFN can also promote pathology (*Barrat and Su, 2019*). While in viral infections, type I IFN generally appears as an absolute requirement for effective antiviral immune responses, in bacterial (*Mycobacterium tuberculosis*, *Listeria monocytogenes*) and parasitic (*Plasmodium*) infections, type I IFN can favor deleterious outcomes (*Auerbuch et al., 2004*; *Spaulding et al., 2016*; *Sharma et al., 2011*; *Donovan et al., 2017*; *Mooney et al., 2017*). This is often associated with the production of an inappropriate set of inflammatory mediators that hampers or skews the development of an effective host response and promotes damaging inflammation, as also exemplified during SARS-CoV-2 infection (*Tay et al., 2020*).

Consistent with the described roles of type I IFN, pDC-derived type I IFN has been reported to be essential for host resistance against a range of viral infections (*Reizis, 2019*; *Barrat and Su, 2019*). It augments antiviral defense through the rapid activation of innate immune effector responses such as that of NK cells, monocytes, and neutrophils (*Cervantes-Barragan et al., 2012*; *Cervantes-Barragan et al., 2007*; *Swiecki et al., 2013*; *Webster et al., 2018*; *Davidson et al., 2011*; *Lynch et al., 2018*). It also modulates long-term CD8[+] and CD4[+] T cell survival and differentiation (*Swiecki et al., 2010*). pDC-derived type I IFN also promotes autoimmunity in SLE and dermatitis, systemic sclerosis, and type I diabetes (*Barrat and Su, 2019*; *Hansen et al., 2015*; *Ah Kioon et al., 2018*; *Guiducci et al., 2010*). Similarly, in a mouse model of severe lethal malaria, pDC-derived type I IFN can drive innate immune cell activation and robust inflammatory responses (*Spaulding et al., 2016*).

Several mechanisms are implicated in pDC initiation of type I IFN production in vivo. These include TLR7/9 pDC-intrinsic sensing of viral nucleic acids occurring as a result of direct pDC infection by viruses (*Lee et al., 2007*; *Décembre et al., 2014*; *Macal et al., 2012*; *Hornung et al., 2004*). Indirect sensing of pathogen-derived materials from non-pDC-infected cells also occurs via exosomes, shedding of immature virus-containing particles or even internalization of immune complexes (*Webster et al., 2016*; *Webster et al., 2018*; *Décembre et al., 2014*; *Dreux et al., 2012*; *Takahashi et al., 2010*; *Björck et al., 2008*; *Rönnblom and Alm, 2001*). pDC may therefore interact with other cells or cell-derived materials, for effective pathogen sensing and subsequent initiation of type I IFN production. In line with this concept, several studies have reported that pDC forms rapid clusters upon TLR7/9-mediated activation (*Asselin-Paturel et al., 2005*; *Mittelbrunn et al., 2009*; *Brewitz et al., 2017*). Systemic injection of synthetic TLR7/9 ligands or murine CMV infection induces pDC to migrate via CXCR3 and CCR7 to the marginal zone of spleens where they achieve peak production of type I IFN (*Asselin-Paturel et al., 2005*). pDC can also form stable immunological synapses with CD4[+] T cells in vitro and upon TLR9-induced activation in the marginal zone/red pulp area (*Asselin-Paturel et al., 2005*). More recently, lymph node (LN) pDC behavior monitored during viral infection showed that they were recruited via CXCR3 or CCR5, respectively, by infected subcapsular CD169[+] MP or activated CD8[+] T cells to the site of priming and interaction with XCR1[+] DC (*Brewitz et al., 2017*). Here, type I IFN from pDC promoted (i) local macrophage (MP)-antiviral response preventing virus spreading and more efficient antigen cross-presentation by DC to CD8[+] T cells. Consistent with these findings, the formation of a dynamic TLR7-dependent interferogenic synapse between pDC- and virus-infected cells was also described in vitro, in which viral RNA and type I IFN were exchanged as possible mechanisms potentiating pDC scanning of infected cells while providing local and targeted delivery of type I IFN (*Assil et al., 2019*). These results therefore suggested that tissue trafficking and localization of pDC in situ is essential for them to undergo activation and for their optimal function.

However, the precise spatiotemporal dynamics of pDC behavior in vivo at steady state and during infection or inflammation, and which cells and molecular events may regulate these events still remain poorly understood. Whether pDC initiation of type I IFN production is linked to their motile behavior in vivo is also largely uncharacterized.

We have previously established that (1) CD169+ MP prime pDC to initiate rapid type I IFN production in the bone marrow (BM) of mice infected with lethal *Plasmodium yoelii* (*Py*) YM (*Spaulding et al., 2016*), (2) pDC priming involved pDC-intrinsic TLR7 and pDC-extrinsic STING sensing, and (3) pDC arrested in the BM of infected mice at the time of peak type I IFN production (~1.5 days) using intravital microscopy (IVM) while they were highly motile in the BM of naive mice. pDC arrested near or potentially in contact with CD169+ MP, however, these MPs were also densely distributed throughout the BM making it difficult to establish a functional interaction. CD169+ MP are tissue-resident MP, which localizes at entry ports of lymphatics in dLNs and in splenic marginal zones. They capture and sample antigens from dead cells, and act as entry points for viruses and bacteria, subsequently initiating immune defenses (*Sung et al., 2012*; *Kastenmüller et al., 2012*; *Perez et al., 2017*; *Miyake et al., 2007*). Building on our prior work, we directly tested the hypothesis that pDCs need to arrest and communicate with CD169+ MP in order to properly activate and initiate type I IFN production. We assess if CD169+ MP take up *Py*-infected red blood cells (iRBCs), acting as a primary source of parasite-derived materials and signals for pDCs. We also explore the molecular sensing mechanisms underlying the establishment of this functional interaction. Our results show that both CD169+ MP and TLR7 sensing in pDC control pDC arrest. We also establish that STING sensing in CD169+ MP is necessary to prime pDC production of systemic type I IFN, and to allow pDC egress from the BM.

## Results

### CD169+ MP take up *P. yoelii*-iRBC in the BM

Since CD169+ MP are required to activate pDC in the BM of blood stage malaria-infected mice (*Spaulding et al., 2016*), we hypothesized that CD169+ MP initiate pDC activation after taking up *Py*-iRBC. Flow cytometric analysis revealed the presence of two subpopulations of CD169+ MP in the BM, which we defined phenotypically as F4/80hi CD169int (F4/80hi MP subset) and F4/80int CD169hi (CD169hi MP subset) (*Figure 1—figure supplement 1A*). To explore uptake of *Py*-iRBC by BM MP, we isolated RBC either from naive (uninfected RBCs, uRBC) or *Py*-infected mice, and labeled them with the red fluorescent dye tetramethylrhodamine (TAMRA-Red+) before transfer to WT recipient mice for FACS(-Fluorescence Activated Cell Sorting) and IVM analyses of the BM (*Figure 1—figure supplement 1B*). This strategy enabled quantification and visualization of the initial uptake phase of *Py*-iRBC versus uRBCs in the BM. For in vivo visualization of CD169+ MP by IVM, mice were injected i.v. with anti-CD169-FITC (or CD169-PE in other experiments) mAb 16 hr prior to IVM imaging (*Spaulding et al., 2016*). This allowed to identify CD169+ MP in the BM as FITChi or PEhi cells, and labeled the same cell subsets as when the Cre recombinase is expressed under the promoter of the gene encoding for CD169 (*Siglec1Cre*) crossed to conditional Rosa26-lox-STOP-lox (LSL)-tomato or -YFP mice (*Figure 1—figure supplement 1C*). As early as 6 hr post-transfer of TAMRA-Red+ RBC, both F4/80hi and CD169hi BM MP took up significantly increased proportions of *Py*-infected iRBCs compared to uRBCs, with a ~10-fold higher uptake of infected relative to uninfected RBC in both CD169+ MP subsets (*Figure 1A*). Of note, we did not detect direct uptake of *Py*-iRBCs by pDC (*Figure 1—figure supplement 2A*). *Py*-iRBC uptake was also comparable in mice injected with anti-CD169 mAb or control isotype Ab prior to the transfer of TAMRA-Red+ iRBC, ruling out that the in vivo labeling of CD169+ MP interfered with *Py*-iRBC uptake (*Figure 1—figure supplement 2B*). Time-lapse IVM imaging confirmed *Py*-iRBC attachment and accumulation on CD169+ MP in the BM, and possibly internalization (*Figure 1B* and *Figure 1—video 1*). Thus, taken together these data show that both populations of CD169+ MP take up *Py*-iRBC in the BM of infected mice in vivo, consistent with the hypothesis that MP can present parasite-derived materials and provide activating signals to pDC.

### CD169+ MP undergo massive loss and activation in *Py*-infected BM

If uptake of *Py*-iRBCs is indeed important for initial sensing of parasite-derived molecular patterns by MPs, we hypothesized that their activation status should be significantly modulated. Thus, we monitored both F4/80hi and CD169hi MP numbers and activation using various cell-surface markers

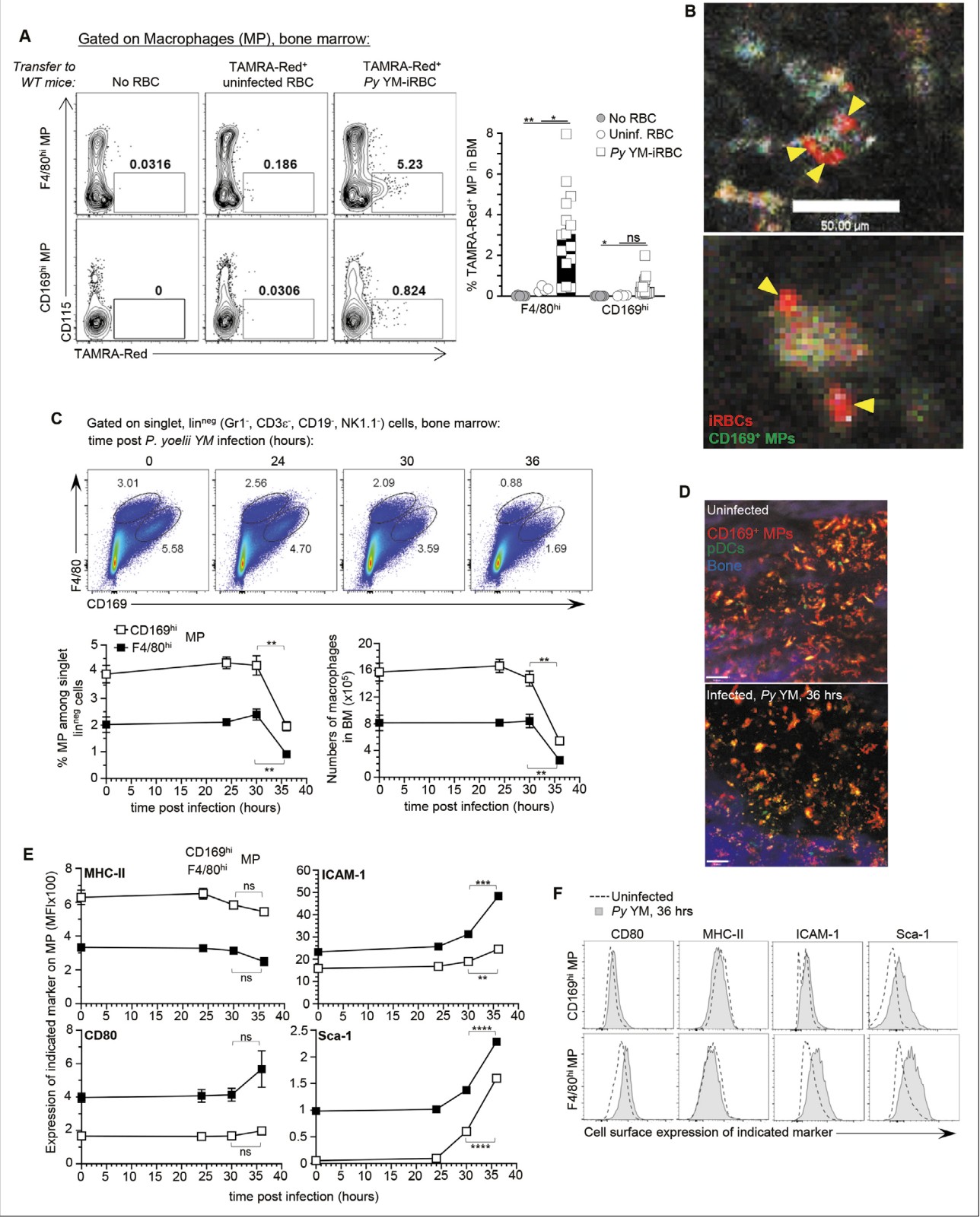

**Figure 1.** CD169+ MP uptake *Plasmodium yoelii*-infected RBC (iRBC) in the bone marrow and undergo activation and loss. F4/80hi and CD169hi MP were gated as depicted in *Figure 1—figure supplement 1A*. (A) Representative dot plots of frequency of TAMRA-Red+ F4/80hi or CD169hi MP from control (no transfer), uninfected RBC (uRBC), and infected RBC (iRBC) transferred recipient mice are shown. Bar graphs show the mean frequencies ± standard error of the mean (SEM) of TAMRA-Red+ MP subsets in a pool of six independent replicate experiments with each symbol representing one mouse (*n*

*Figure 1 continued*

= 4–15 mice/condition). (**B**) Intravital image of TAMRA-Red⁺ *Py*-iRBC (red) associated with anti-CD169-FITC labeled CD169⁺ MP (green) in tibial BM 6 hr post-transfer. Sixteen hours prior to imaging, mice were i.v. administered CD169-FITC mAb to label CD169⁺ MP. Yellow arrowheads show *Py*-iRBC that are associated with CD169⁺ MP. Images are representative of two replicate experiments. (**C, E, F**) BM cells from uninfected or *Py* YM-infected (5 × 10⁵ *Py*-iRBC) WT mice were harvested at indicated time points and stained with cell-surface expression of linⁿᵉᵍ (CD3, CD19, NK1.1, CD19), F4/80, CD169, and several activation markers (MHC-II, ICAM-1, CD80, Sca-1). In (**C**), representative FACS dot plots of CD169ʰⁱ and F4/80ʰⁱ MP in the BM. The relative proportion and absolute numbers CD169ʰⁱ and F4/80ʰⁱ MP in mouse leg of WT B6 mice are shown in a pool of two replicate experiments with SEM (*n* = 3–7 mice for each time point). (**D**) Representative image of tibial BM of *PTCRA*-EGFP mice administered with CD169-PE mAb 16 hr prior to imaging either uninfected or 36 hr post *Py* infection in 1 of >10 imaged BM. (**E, F**) Representative overlay FACS histograms for indicated activation marker on CD169ʰⁱ and F4/80ʰⁱ MP from indicated mice and conditions. When relevant, p values were calculated with *p < 0.05; **p < 0.01; ***p < 0.001; ****p < 0.0001; ns, not significant, using two-tailed unpaired Student's *t*-test.

The online version of this article includes the following video, source data, and figure supplement(s) for figure 1:

**Source data 1.** Raw values for *Figure 1A, C, E* graphs.

**Figure supplement 1.** Gating strategies and experimental design for *Figure 1*.

**Figure supplement 2.** Analysis of pDC and CD169 roles in uptake of Py-iRBC, and CD169+ MP activation in infected mice.

**Figure supplement 2—source data 1.** Raw values for for *Figure 1—figure supplement 2A-C* graphs.

**Figure 1—video 1.** Visualization of *Py* iRBCs uptake by CD169⁺ MP in the bone marrow of living mice.

https://elifesciences.org/articles/78873/figures#fig1video1

related to type I IFN signaling (Sca-1), antigen presentation (MHC-II, CD80, CD86), and adhesion/uptake (ICAM-1, CD11a) following *Py* infection (*Figure 1C–F* and *Figure 1—figure supplement 2C*). We found a significant loss (>50%) of both MP subsets in the BM of *Py*-infected mice, that occurred between 30 and 36 hr post-infection as quantified by FACS analysis (*Figure 1C*) and IVM imaging of the tibial BM where the decrease in the overall BM MP numbers was also visualized in situ (*Figure 1D*). In addition, Sca-1, CD80 costimulatory molecule, and ICAM-1 integrin ligand expression were increased while CD11a was diminished, consistent with robust activation of both subsets of MP in response to *Py* infection (*Figure 1E, F* and *Figure 1—figure supplement 1D*). In summary, CD169⁺ MP in the BM of *Py*-infected mice are rapidly lost as they undergo robust activation.

## CD169⁺ MP are required for pDC arrest in *Py*-infected BM

To directly link CD169⁺ MP priming of pDC to pDC motile behaviors, we next assessed whether *Py*-activated after *Py* infection control pDC arrest using time-lapse IVM of the tibial marrow. As an initial step, we established baseline pDC motile behaviors using several readouts comparing uninfected versus *Py*-infected WT mice at the time pDC produce peak type I IFN (i.e., 36 hr). We took advantage of the *PTCRA*-EGFP reporter mice, that express the enhanced green fluorescent protein (EGFP) under the control of the human pre-T cell receptor α (*PTCRA*) promoter. This promoter is selectively expressed in BM pDC (>95% of GFP⁺ cells, *Figure 2—figure supplement 1A*, *Shigematsu et al., 2004*) enabling us to visualize pDCs in vivo. For CD169⁺ MP visualization, mice were injected intravenously with the anti-CD169-PE mAb prior to IVM imaging (*Figure 1—figure supplement 1C*). To quantify pDC dynamic behavior, we utilized several metrics, which include track velocity (TV) that measures cell path length over time, displacement rate (DR) that measures overall distance cells moved (irrespective of path) over time, and mean square displacement (MSD) that classifies cell movement as random, confined, or processive and inform on the cell behavior. In WT uninfected mice, pDC was motile, exhibiting long tracks and a TV of 7.14 µm/min (*Figure 2A, B* and *Figure 2—video 1*), faster than their reported speed of 5 µm/min in LNs (*Mittelbrunn et al., 2009*), but slower than naive T cells in the LN that travel at 10–15 µm/min (*Shigematsu et al., 2004*). Based on their DR (1.5 µm/min) and MSD, pDC migrated through and explored a large volume of parenchymal tissue (~1350 µm²/hr), similar to LN lymphocytes (*Mittelbrunn et al., 2009*) and consistent with a scanning role in the BM (*Figure 2B–D*). In contrast, CD169⁺ MP were sessile, whether in naive or infected mice (*Figure 2—figure supplement 1B*). After *Py* infection, pDC tracks, TV (5.13 µm/min), DR (0.93 µm/min), and MSD were all significantly reduced. These quantitative measures are also consistent with their arrest visualized by IVM. Average mean values of these distinct measurements from independent mice reflected these findings. We next tested whether CD169⁺ MP were involved in pDC arrest after *Py* infection. We used (1) *PTCRA*-EGFP reporter mice crossed to *siglec1ᴰᵀᴿ/ᴰᵀᴿ* mice and (2) sublethally

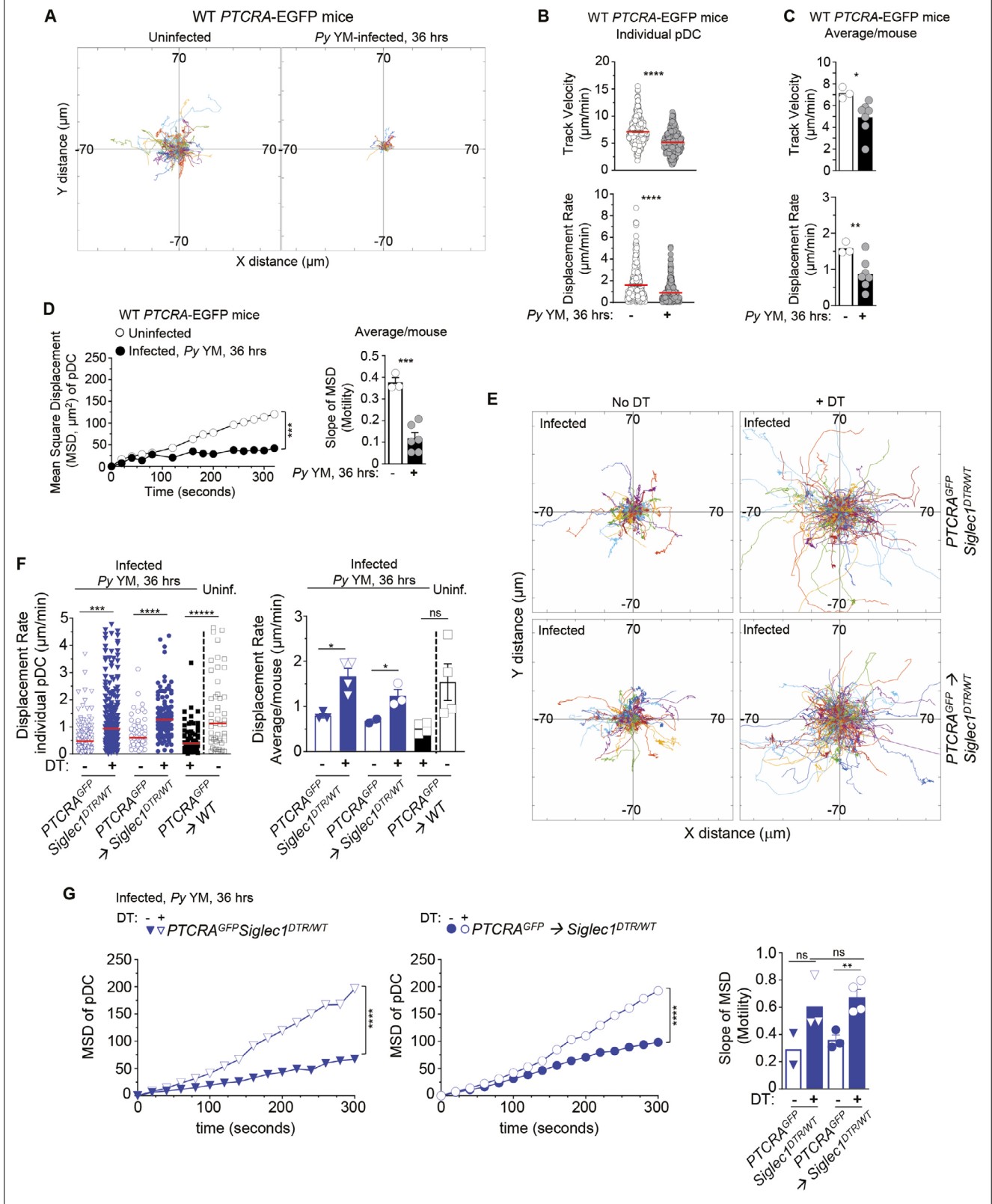

**Figure 2.** CD169+ MP control pDC arrest in *Py*-infected mice. *PTCRA*-EGFP reporter mice (WT, *siglec1^DTR/WT*, or partial bone marrow chimeras) either received PBS (uninfected) or were infected with 5 × 10^5 *Py*-iRBCs, and administered with CD169-PE mAb 16 hr prior to intravital imaging microscopy (IVM) of tibial BM. (**A–D**) Analysis of pDC dynamic behavior in uninfected or 36 hr *Py*-infected WT *PTCRA*-EGFP mice. (**A**) Flower plots of 2D tracks of individual pDCs, superimposed after normalization of their starting coordinates to the origin over 30 min of imaging. (**B**) Track velocity (TV) and

*Figure 2 continued on next page*

*Figure 2 continued*

displacement rates (DRs) of individual pDC with (**C**) average in all mice, and (**D**) mean square displacement (MSD) analysis of pDC over time and slope of MSD/motility. For bar graphs, each symbol features one mouse with mean ± standard error of the mean (SEM) (*n* = 3–7 mice were imaged for each experimental condition and genotype). (**E–G**) *PTCRA*-EGFP *siglec1*$^{DTR/WT}$ or *PTCRA*-EGFP/*siglec1*$^{DTR/WT}$ and *PTCRA*-EGFP/WT partial BM chimeras were injected with diphtheria toxin (DT) or control PBS 12 hr prior to *Py* infection. (**E**) Tracks of individual pDC in the indicated experimental conditions over 1 hr of imaging. (**F**) DRs of individual pDC and average in all mice. (**G**) MSD analysis of pDC over time and slope of MSD/motility. For all bar graphs, each symbol features one mouse with mean ± SEM (*n* = 2–4 mice were imaged for each experimental condition and genotype). p values were calculated with *p < 0.05; **p < 0.01; ***p < 0.001; ****p < 0.0001; ns, not significant, using two-tailed unpaired Student's *t*-test for flow-cytometry comparisons and Welch's *t*-test for comparison of IVM quantifications. Multiple linear regression analyses were applied for statistical analysis of the MSD plots.

The online version of this article includes the following video, source data, and figure supplement(s) for figure 2:

**Source data 1.** Raw values for *Figure 2A-G* data.

**Figure supplement 1.** Validation of reporter mouse tools.

**Figure supplement 1—source data 1.** Raw values for *Figure 2—figure supplement 1B, C* graphs/data.

**Figure 2—video 1.** *Py* infection induces pDC arrest.

https://elifesciences.org/articles/78873/figures#fig2video1

**Figure 2—video 2.** CD169$^+$ MP depletion prevents pDC arrest during *Py* infection.

https://elifesciences.org/articles/78873/figures#fig2video2

irradiated *siglec1*$^{DTR/WT}$ or control WT recipient mice (500 rads) reconstituted with WT *PTCRA*-EGFP reporter BM cells. These chimeras allowed us to rule out any substantial impact of the DTR transgene in pDC since WT *PTCRA*-EGFP$^+$ pDC did not express the transgene. In *siglec1*$^{DTR}$ mice, whether chimeric or not, diphtheria toxin (DT) injection induced the selective depletion of both subsets of CD169$^+$ MP (*Figure 2—figure supplement 1C*). Interestingly, IVM imaging of *Py*-infected DT-treated mice revealed much longer pDC tracks compared to untreated control mice (*Figure 2E*). Moreover, pDC DR and MSD were significantly increased compared to untreated mice (by ~50%, *Figure 2F, G* and *Figure 2—video 2*). As expected, in *Py*-infected, DT-treated *PTCRA*-EGFP/WT control chimeras, pDC had significantly reduced motility compared to uninfected. Average mean values from independent mice were also consistent with these conclusions. Thus, these data collectively establish that CD169$^+$ MP control pDC dynamics during *Py* infection and suggest that pDC arrest is linked to their ability to initiate type I IFN production.

## pDC arrest in *Py*-infected BM depends on TLR7-intrinsic signaling

To achieve type I IFN production, pDC requires TLR7-intrinsic sensing (*Spaulding et al., 2016*). If pDC arrest is indeed linked to their robust activation, we predicted that TLR7 sensing in pDC would also be required for their arrest. We first generated *Tlr7*$^{-/y}$ and WT *PTCRA*-EGFP reporter mice and monitored pDC motility in the tibial BM of *Py* infected (36 hr post-infection), or control uninfected live mice using IVM imaging. In both *Tlr7*$^{-/y}$ and WT uninfected mice, pDC was highly motile with long tracks and a comparable TV (respectively, 6.52 and 7.16 μm/min, *Figure 3A, B* and *Figure 3—video 1*). Interestingly, pDC in *Tlr7*$^{-/y}$ uninfected mice exhibit greater motility, suggesting that tonic TLR7 signals regulate pDC motility under steady-state conditions (*Figure 3D*). Following *Py* infection in WT mice, pDC motility was reduced by 31%, in terms of TV (4.92 μm/min) and 45% based on DR (~0.87 μm/min), as well as for MSD and slope of MSD averages (*Figure 3B, D* and *Figure 3—video 2*). In contrast, pDC in *Py*-infected *Tlr7*$^{-/y}$ mice did not arrest and continued migrating with similar speeds in terms of TV (7.05 μm/min) and DR (1.55 μm/min, *Figure 3B, C*) despite detectable measurable MSD reductions (*Figure 3D*) reflecting a more restricted migration pattern, but not arrest. Average mean values from independent mice also largely confirmed these conclusions, overall suggesting that while other factors than TLR7 signals control pDC motility, these remain insufficient to completely arrest them. Representing the proportion of pDC moving or arrested based on WT pDC threshold values (both for TV and DR) further highlights a comparable proportion of pDC moving or arrested in *Py*-infected *Tlr7*$^{-/y}$ mice and naive WT or *Tlr7*$^{-/y}$ mice, in contrast to *Py*-infected WT counterpart (*Figure 3—figure supplement 1A*). To further assess if cell-intrinsic TLR7 sensing mediated pDC arrest, we generated partial mixed BM chimera mice as above, in which WT recipient mice (CD45.1$^+$) were grafted with either *Tlr7*$^{-/y}$- or WT-*PTCRA*-EGFP reporter BM after sublethal irradiation. In these mice, ~30% of hematopoietic-derived cells originated from the donor reporter BM, allowing for tracking of *Tlr7*$^{-/y}$

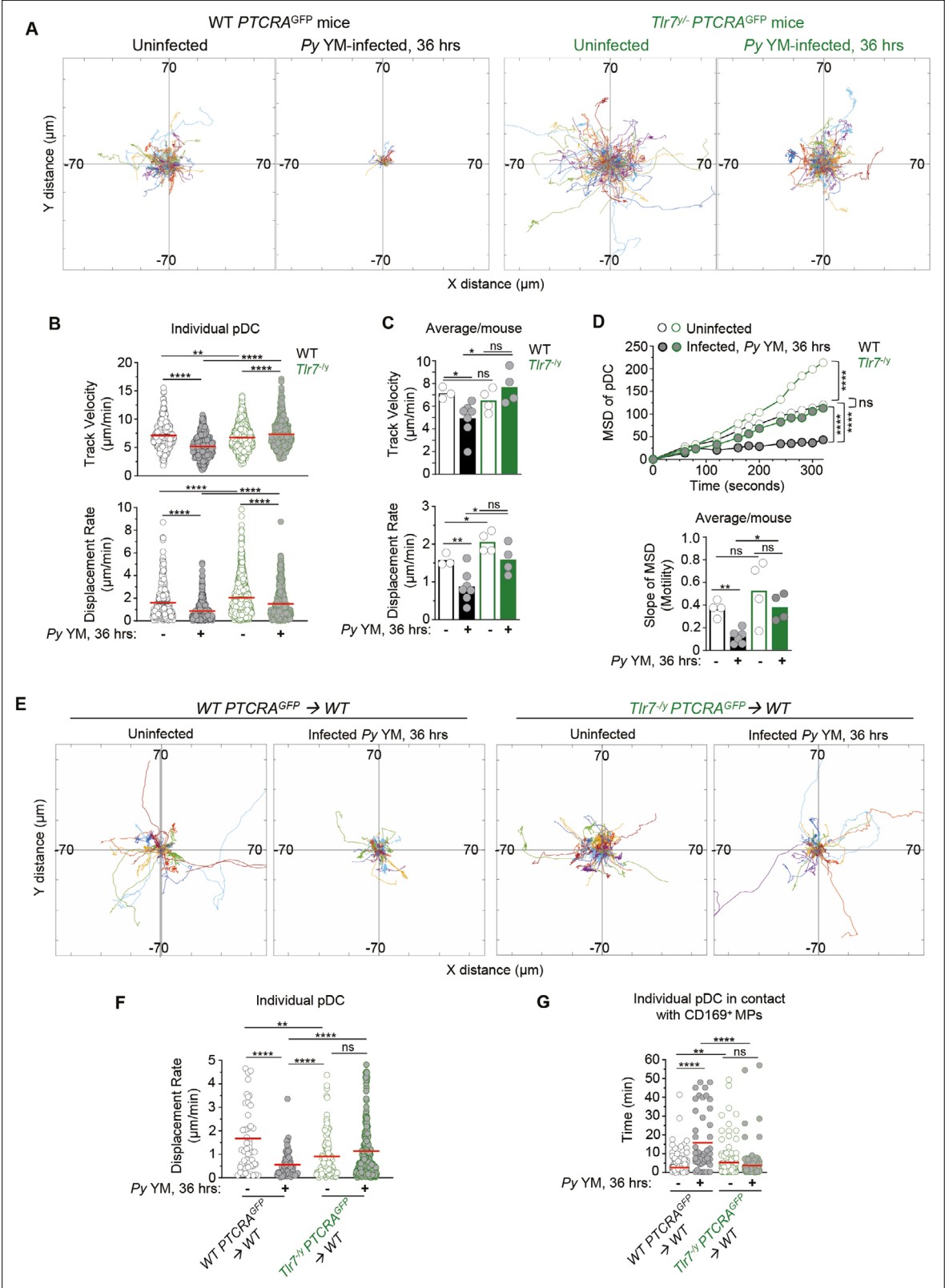

**Figure 3.** TLR7 sensing in pDC is required for their arrest and clustering in *Py*-infected mice. WT and *Tlr7⁻/y PTCRA*-EGFP reporter mice, and WT *PTCRA*-EGFP/WT and *Tlr7⁻/y PTCRA*-EGFP/WT partial bone marrow (BM) chimeras either received PBS (uninfected) or were infected with 5 × 10⁵ *Py*-iRBCs, and administered with CD169-PE mAb 16 hr prior to intravital microscopy (IVM) of tibial BM. (**A**) Tracks of individual pDC dynamic behavior in uninfected or 36 hr *Py*-infected mice with starting positions in the same origin over 30 min of imaging. (**B**) Track velocity (TV) and displacement rates

*Figure 3 continued on next page*

*Figure 3 continued*

(DRs) of individual pDC with (**C**) average in all mice, and (**D**) mean square displacement (MSD) analysis of pDC over time and slope of MSD/motility. Each symbol feature one mouse with mean ± standard error of the mean (SEM). Four to six individual mice were imaged for each experimental condition and genotype. (**E**) Tracks of individual pDC in uninfected or *Py*-infected WT *PTCRA*-EGFP/WT and *Tlr7⁻/ʸ PTCRA*-EGFP/WT partial BM chimeras over 1 hr of imaging. (**F**) DR of individual pDC and (**G**) quantification of individual pDC contact time with CD169⁺ MP are shown. Three to four individual mice were imaged for each experimental condition and genotype. p values were calculated with *p < 0.05; **p < 0.01; ; ****p < 0.0001; ns, not significant, using two-tailed unpaired Student's *t*-test for flow-cytometry comparisons and Welch's *t*-test for comparison of IVM quantifications. Multiple linear regression analyses were applied for statistical analysis of the MSD plots. The WT PTCRA-EGFP data depicted in (**A**) are the same as in *Figure 2A*; (**B, C**) are identical as in *Figure 2B, C*; (**D**) are the same as in *Figure 2D*; (**F**) are the same as in *Figure 2F*. This enabled comparisons across mutant mouse conditions relative to WT.

The online version of this article includes the following video, source data, and figure supplement(s) for figure 3:

**Source data 1.** Raw values for *Figure 3A-G* graphs/data.

**Figure supplement 1.** Impaired pDC expression of chemokine receptors and adhesion molecules in *Py*-infected TLR7-deficient mice.

**Figure supplement 1—source data 1.** Raw values for *Figure 3—figure supplement 1C* graphs.

**Figure 3—video 1.** pDCs are highly motile in bone marrow of WT and *Tlr7⁻/ʸ* naive mice.

https://elifesciences.org/articles/78873/figures#fig3video1

**Figure 3—video 2.** pDC remains highly motile in the bone marrow (BM) of *Tlr7⁻/ʸ* mice during *Py* YM infection.

https://elifesciences.org/articles/78873/figures#fig3video2

**Figure 3—video 3.** TLR7-deficient pDC fail to arrest in the bone marrow (BM) of *Tlr7⁻/ʸ PTCRA*-EGFP/WT partial BM chimeras during *Py* infection.

https://elifesciences.org/articles/78873/figures#fig3video3

---

or control WT *PTCRA*-EGFP⁺ pDC in an environment where ~70% of HSC-derived cells and all radio-resistant cells, were intact for TLR7 signaling (*Figure 3—figure supplement 1B*). Consistent with the prior result (*Figure 3A–D*), *Tlr7⁻/ʸ* pDC in infected chimeras exhibited long tracks and did not arrest while WT pDC had much shorter tracks and arrested, establishing that TLR7 signaling intrinsic to pDC, is required for them to stop during infection (*Figure 3E, F* and *Figure 3—video 3*). We also quantified pDC/CD169⁺ MP contact durations using a Kiss and Run algorithm (Imaris) as a measure of their possible interactions in chimeric mice and found that WT pDC significantly increased their contact times with the MP during infection (from ~2 to 16 min) whereas *Tlr7⁻/ʸ* pDC interacted with the MP for comparable amounts of time (~4–5 min) whether mice were infected or not (*Figure 3G*). Thus, taken together, these results show that TLR7 sensing in pDC is required for their arrest and potential interactions with CD169⁺ MP in the BM of malaria-infected mice at the time pDC produce peak type I IFN. Since CD169⁺ MP are required for pDC arrest, it further suggests that MP are providing TLR7 ligands to pDC.

## Altered chemotactic receptor and adhesion molecule expression on pDCs from *Py*-infected *Tlr7⁻/ʸ* compared to WT mice

Because TLR7 sensing alters pDC dynamic behavior in vivo, we next hypothesized that TLR7 signaling likely modulated both their migratory and adhesion characteristics. Several chemotactic receptors (CXCR3, CXCR4, CX3CR1, CCR2, CCR5, CCR9) and adhesion molecules (CD11a, ICAM-1, VCAM-1) were reported to be expressed on pDC in naive mice and to contribute to their trafficking and ability to cluster upon activation (*Asselin-Paturel et al., 2005*; *Brewitz et al., 2017*; *Sawai et al., 2013*). We monitored cell-surface expression of chemokine receptors (CXCR3, CCR9, CXCR4, CX3CR1) and integrin ligands (ICAM-1, VCAM-1) on pDC in uninfected compared to *Py*-infected WT and *Tlr7⁻/ʸ* mice (*Figure 3—figure supplement 1C*). While basal expression of these receptors and integrins was comparable in all groups of uninfected mice, CCR9 and CX3CR1 were significantly upregulated on pDC after infection in WT but not in *Tlr7⁻/ʸ* mice. Significant increases in CXCR4 expression were not reliably detectable. In contrast, CXCR3 expression was significantly downregulated on pDC in both WT and *Tlr7⁻/ʸ* mice, but to different extents (90% and 50%, respectively). Expression of ICAM-1, a ligand for the integrin LFA-1, was significantly upregulated in both WT and *Tlr7⁻/ʸ* after infection, but the magnitude of upregulation remained substantially greater in WT mice (factor of ~3). Expression of CD11a, the alpha chain of LFA-1, was slightly decreased in *Py*-infected WT mice whereas expression did not change in *Tlr7⁻/ʸ* mice. Lastly, expression of VCAM-1, ligand for the integrin VLA-4, was significantly reduced in both WT and *Tlr7⁻/ʸ* during the infection, but to a lesser extent in *Tlr7⁻/ʸ* mice.

Collectively, these data link TLR7-mediated sensing by pDCs to changes in cell-surface expression of several chemotactic receptors and adhesion molecules that may contribute to modifications in pDC motility, homing, and/or arrest that occur following *Py* infection.

## $G_{\alpha I}$-protein signaling is required for inducing pDC production of type I IFN

Since we found important modulations in chemokine receptor expression driven by TLR7 signaling, we hypothesized that chemotaxis was involved in pDC motility and arrest. To test this possibility, we injected *Py*-infected mice with either PBS (control) or pertussis toxin (PT), which blocks $G_{\alpha I}$-protein-coupled receptor signal transduction and related chemotaxis, and subsequently analyzed pDC motility, morphology, and type I IFN production (*Figure 4* and *Figure 4—video 1*). In *Py*-infected *PTCRA*-EGFP reporter mice, PT treatment led to stronger immobilization of pDC compared to control mice, with shorter tracks and significantly lower DR (*Figure 4A, B*). pDC arrest was also concomitant with distinct morphological changes (*Figure 4C*). Notably, pDCs in PT-treated mice exhibited a round shape contrasting with the elongated dendritic shape observed in PBS-treated counterparts. Quantitative analysis of individual pDC length, surface, and shape factor (a measure of cell sphericity) further revealed significant differences in PT- versus PBS-treated *Py*-infected mice (*Figure 4D*). While after PT-treatment pDCs were immobile and unable to migrate, they were still viable in the BM and could potentially sense freely diffusing parasite-derived ligands. To determine whether PT treatment impacted the ability of pDC to make type I IFN, we carried out the same experiment in *Ifnb*[YFP/YFP] reporter mice (*Scheu et al., 2008*), which revealed an 80% reduction in type I IFN-expressing pDC (YFP[+]) compared to control infected group (0.44% vs. 2.1% YFP[+] pDCs, *Figure 4E*). Taken together with the imaging findings, this indicated that $G_{\alpha I}$-protein-dependent transduction was required to achieve maximal type I IFN production. Despite their drastic immobilization and morphological changes, pDC in PT-treated infected mice still modulated cell-surface expression of costimulatory (CD86) and adhesion (CD62L, ICAM-1, VCAM-1) molecules, as well as chemotactic receptors (CXCR3), compared to PBS-treated infected and uninfected control mice (*Figure 4E*). Specifically, expression of CD62L, ICAM-1, VCAM-1, and CXCR3 but not CD86, were significantly altered in PT- compared to PBS-treated *Py*-infected mice. These results show that blocking $G_{\alpha I}$-protein-coupled receptor signal transduction including chemotaxis, alters expression of most but not all of these markers, and the acquisition of type I IFN-secreting capacity by the pDC.

The fact that PT treatment prevented pDC motility, and both CD169[+] MP and TLR7 sensing were required for pDC arrest and type I IFN production, suggested that pDCs do not encounter freely circulating TLR7 ligands in the BM but rather that such ligands must be presented by *Py*-activated CD169[+] MP through close and durable contacts with pDC. However, as PT treatment has pleiotropic activity and may induce indirect effects on other cells, we next attempted to determine more specifically which chemotactic and adhesion receptors mediated pDC homing and activation. CXCR3 is significantly downregulated after *Py* YM (*Figure 3—figure supplement 1B*) and has been shown to promote chemotaxis of pDC to sites of priming in the LN (*Brewitz et al., 2017*). CXCR3 has two ligands, CXCL10 and CXCL9, which are highly expressed by monocytes and MP during infections (*Soudja et al., 2014*; *Kastenmüller et al., 2013*). Thus, we tested if CXCR3 was functionally contributing to pDC activation in the BM. We injected *Ifnb*[YFP/YFP] mice at the time of infection both with anti-CXCR3 (CXCR3-173 clone) and anti-CXCL9 2A6.9.9 mAb, which blocks the CXCL10-binding site and neutralizes soluble CXCL9, respectively (*Figure 4F*). pDC in these mice expressed YFP at similar levels relative to control-isotype Ab-treated or non-treated mice (1.5–2%). Furthermore, Ab blocking of LFA-1 and VLA-4 integrins, with or without anti-CXCR3 mAb, also failed to decrease expression of type I IFN by the pDC, suggesting that none of the tested reagents were sufficient to recapitulate the in vivo effects of PT treatment on pDC activation. Collectively, these results suggested that pDC motility requires $G_{\alpha I}$-dependent chemotaxis. This process controls pDC homing to where TLR7 ligands are, and direct or indirect functional interactions with CD169[+] MP to initiate type I IFN production.

## STING signaling is not required for pDC arrest during *Py* infection

Since (1) STING signaling extrinsic to pDC is required for pDC to secrete type I IFN during *Py* YM infection (*Spaulding et al., 2016*), and (2) pDC arrest and possible functional interaction with MP, is linked to their full activation, we hypothesized that STING sensing could also contribute to pDC homing,

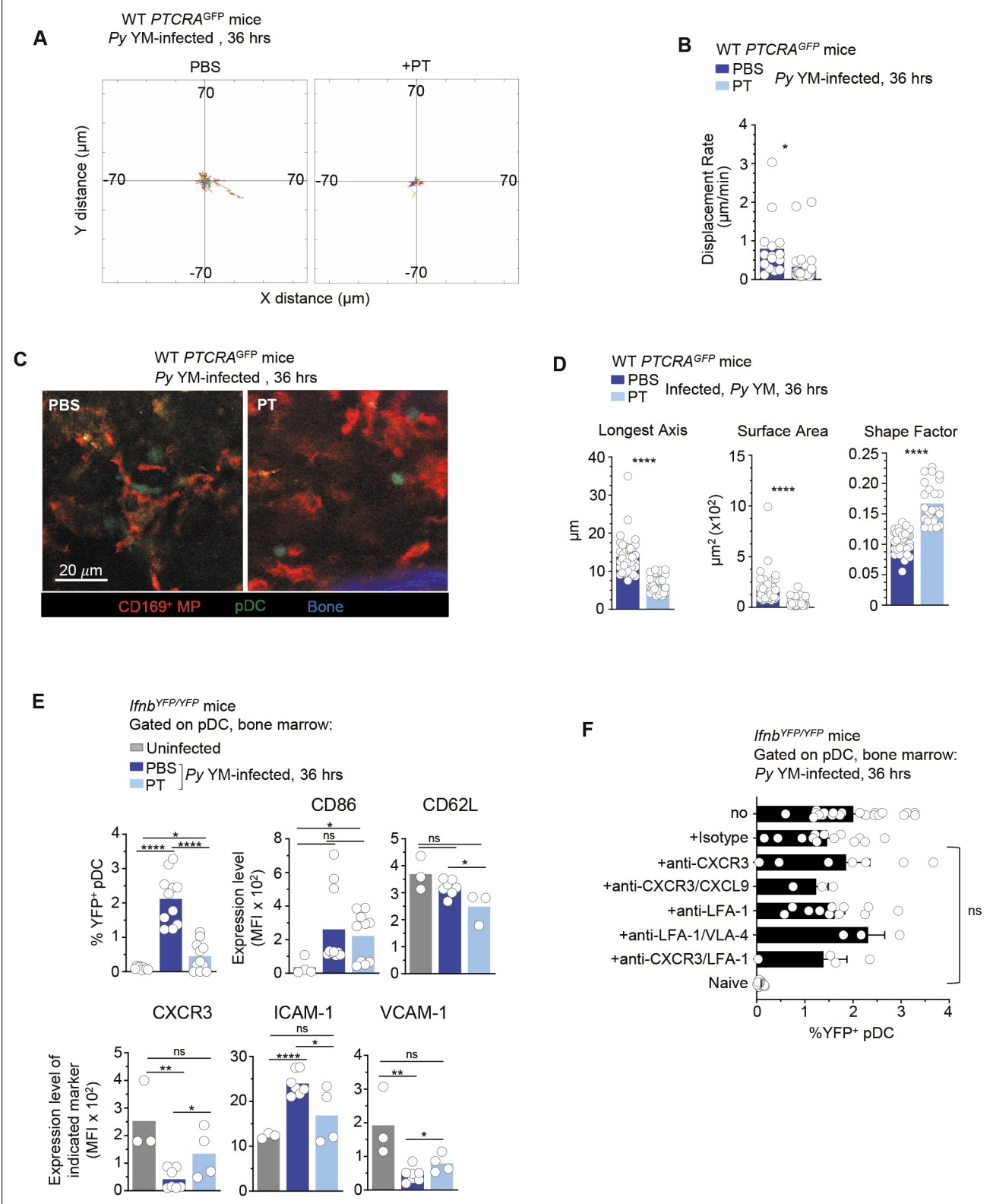

**Figure 4.** pDC arrest is not sufficient to achieve robust type I IFN production during *Py* infection. WT *PTCRA*-EGFP (**A–D**) or *Ifnb*^YFP/YFP^ (**E, F**) reporter mice either received PBS or pertussis toxin (PT) and were infected with 5 × 10⁵ *Py*-iRBCs. In (**A–D**), mice were injected with PT or control PBS at the time of *Py* infection and administered with CD169-PE mAb 16 hr prior to intravital microscopy (IVM) of tibial bone marrow (BM). The analysis of pDC dynamic behavior of PT-treated or untreated, 36 hr *Py*-infected mice are shown. (**A**) Flower plots of 2D tracks of individual pDCs, superimposed after

Figure 4 continued

normalization of their starting coordinates to the origin over 30 min of imaging. (**B**) Displacement rate (DR) of individual pDC. (**C**) Representative IVM images of tibial BM. (**D**) Quantification of cell morphology of arrested pDC in PBS versus PT-treated infected mice. (**E, F**) $Ifnb^{YFP/YFP}$ reporter mice received indicated treatments (PBS, PT, Isotype Ab, anti-CXCR3+/−CXCL9+/−LFA-1, anti-LFA-1+/−VLA-4) and the time of $Py$ infection, and 36 hr later BM cells were stained for cell-surface expression of lin$^{neg}$ (CD3, CD19, NK1.1, CD11b, Gr1), BST-2, Siglec-H, CD86, CD62L, CXCR3, VCAM-1, and ICAM-1. Bar graphs quantify the expression of YFP, CD86, and ICAM-1 by pDCs with mean ± standard error of the mean (SEM) across two to five independent replicate experiments, with each symbol featuring one mouse ($n$ = 3–10 mice). p values were calculated with *p < 0.05; **p < 0.01; ****p < 0.0001; ns, not significant, using two-tailed unpaired Student's $t$-test for flow-cytometry comparisons and Welch's $t$-test for comparison of IVM quantifications.

The online version of this article includes the following video and source data for figure 4:

**Source data 1.** Raw values for *Figure 4A, B, D-F* graphs/data.

**Figure 4—video 1.** Blocking $G_{\alpha i}$ signaling induces robust pDC arrest and a morphologically distinct phenotype.
https://elifesciences.org/articles/78873/figures#fig4video1

arrest and communication with MP to initiate type I IFN production. To investigate these possibilities, we analyzed pDC motility in $Sting1^{Gt/Gt}$ PTCRA-EGFP reporter mice by IVM. In uninfected $Sting1^{Gt/Gt}$ mice, most pDC exhibited high motility while they arrested in $Py$-infected counterparts (*Figure 5A* and *Figure 5—video 1*). The majority of pDC tracks were confined and with reduced displacement from origin following the infection. While pDC DR and MSD slopes were significantly diminished compared to naive mice, the average pDC TV remained elevated (6.52 µm/min), similar to that of uninfected mice (6.33 µm/min) (*Figure 5B–D*). Such discordance between TV and DR usually means that cells exhibit a more confined migration pattern. Consistent with this interpretation, the MSD of pDC was significantly higher in $Sting1^{Gt/Gt}$ compared to WT-infected mice (*Figure 5D*). We next infected partial BM chimeras in which sublethally irradiated $Sting1^{Gt/Gt}$ or control WT mice were reconstituted with WT PTCRA-EGFP donor BM, allowing to monitor WT pDC motility in a $Sting1^{Gt/Gt}$ host where the majority of HSC-derived cells (~70%) and all radioresistant cells are deficient for STING. In these mice where we visualized WT pDC in a $Sting1^{Gt/Gt}$ host, pDC arrested and had longer contact durations with MP compared to control WT chimera and non-chimera mice (*Figure 5E, F* and *Figure 5—video 2*). Most interestingly, we observed that in both non-chimera $Py$-infected $Sting1^{Gt/Gt}$ mice, pDC clusters were significantly larger in volume than in WT control mice, by threefold on average, and in some cases exceeding WT clusters by tenfold (*Figure 5G, H* and *Figure 5—videos 2 and 3*). We also enumerated by FACS analysis higher numbers of pDC in the BM of $Sting1^{Gt/Gt}$ compared to WT $Ifnb^{YFP/YFP}$ reporter $Py$-infected mice (*Figure 5I*), correlating with the significantly larger clusters of pDC quantified in the BM of $Sting1^{Gt/Gt}$ mice. As expected, the number of $Ifnb$-expressing pDC (YFP⁺) was significantly reduced compared to WT counterparts, despite increased pDC numbers. In addition, while still significantly modulated, expression of several chemotactic receptors (CXCR3, CCR9, CXCR4, CX3CR1) and the integrin VCAM-1, failed to undergo as much cell-surface modulation as in WT mice (*Figure 5—figure supplement 1*). This suggested that pDC in $Sting1^{Gt/Gt}$ mice may not receive appropriate signals to enable them to resume motility, leave clusters and egress from the BM. Taken together, these data establish that pDC-extrinsic, STING-dependent signals do not control pDC homing and arrest in the BM during $Py$ infection, but that they may control pDC functional interactions with CD169⁺ MP and egress from the BM, thereby limiting their accumulation and/or retention in the BM.

## STING signaling in CD169⁺ MP controls pDC production of type I IFN and egress from the BM

Next, we assessed if STING signaling specifically in CD169⁺ MP during $Py$ YM infection controlled (1) initiation of type I IFN production by pDC, and (2) pDC accumulation versus egress from the BM. We bred $siglec1^{Cre/Cre}$ knock-in mice to $Sting1^{Gt/Gt}$ or $Sting1^{F/F}$ mice (where exons 3–5 of $Sting1$ were flanked with two LoxP sites) to generate $siglec1^{Cre/WT}Sting1^{Gt/F}$ mice, in which CD169⁺ MP lacked expression of functional STING. All these mice were also homozygous for the $Ifnb^{YFP}$ reporter transgene, allowing us to monitor type I IFN expression by FACS. This reporter faithfully tracks with the production of IFNα and IFNβ cytokines as we established before (*Spaulding et al., 2016*). Upon $Py$ YM infection, while ~2.2% of pDC (~4200 pDC/leg) in WT and $Sting11^{Gt/WT}$ control mice expressed IFNβ (YFP⁺ ICAM-1$^{hi}$), we quantified a ~60% reduction in YFP⁺ pDC when STING was lacking in CD169⁺ MPs (*Figure 6A*), demonstrating that STING signaling in CD169⁺ MP was required for pDC priming of type I IFN producing capacity. Other markers strongly modulated on pDC upon activation (ICAM-1,

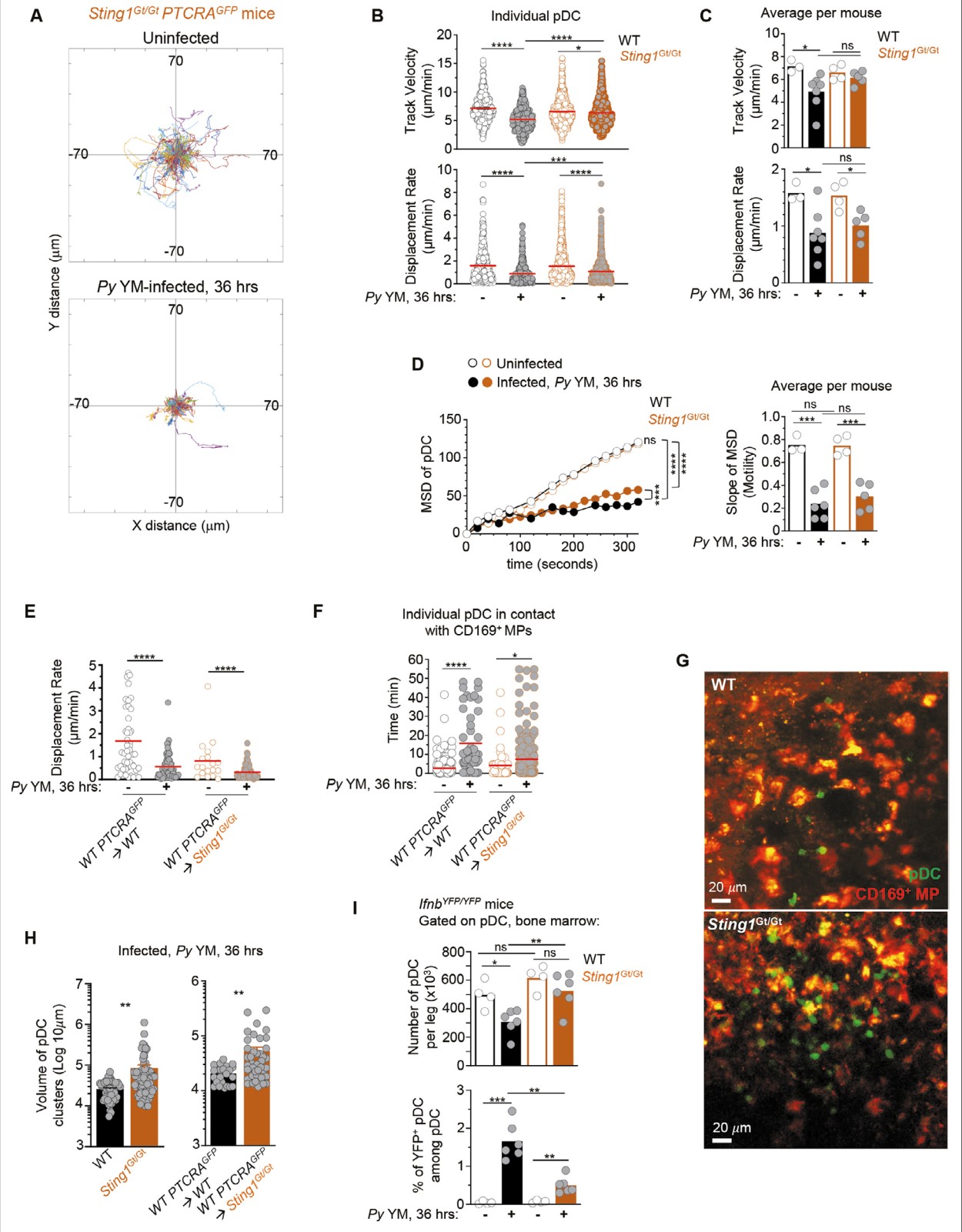

**Figure 5.** pDCs arrest, form larger clusters and accumulate in the bone marrow of STING-deficient compared to WT *Py*-infected mice. WT or *Sting1*<sup>Gt/Gt</sup> *PTCRA*-EGFP reporter mice either received PBS (uninfected) or were infected with 5 × 10⁵ *Py*-iRBCs, and administered with CD169-PE mAb 16 hr prior to intravital microscopy (IVM) imaging of tibial BM. The analysis of pDC dynamic behavior of uninfected or 36 hr *Py*-infected mice is shown. (**A**) Tracks of individual pDC in *Sting1*<sup>Gt/Gt</sup> *PTCRA*-EGFP reporter mice with starting positions in the same origin over 30 min of imaging. (**B**) Track velocity

*Figure 5 continued on next page*

*Figure 5 continued*

(TV) and displacement rate (DR) of individual pDC with (**C**) average in all mice, and (**D**) mean square displacement (MSD) analysis of pDC over time and slope of MSD/motility. Each symbol features one mouse with mean ± standard error of the mean (SEM). Three to seven individual mice were imaged for each experimental condition and genotype. (**E**) DR of individual pDC and (**F**) individual pDC contact time with CD169$^+$ MP, in uninfected or *Py*-infected *Sting1$^{Gt/Gt}$ PTCRA*-EGFP/WT and WT *PTCRA*-EGFP/WT partial BM chimeras. (**H**) Quantification of cluster volume in WT, *Sting1$^{Gt/Gt}$*, *Sting1$^{Gt/Gt}$ PTCRA*-EGFP/WT, and WT *PTCRA*-EGFP/WT partial BM chimeras. (**G**) Image from IVM of tibial BM showing pDC clusters 36 hr post *Py* infection. For (**E–H**), three to four individual mice were imaged for each experimental condition and genotype. (**I**) Number of pDC per leg and relative proportion of YFP$^+$ pDC in the BM of uninfected or *Py*-infected (36 hr) WT- and *Sting1$^{Gt/Gt}$-Ifnb$^{YFP/YFP}$* reporter mice, as quantified by FACS after staining for cell-surface expression of lin$^{neg}$ (CD3, CD19, NK1.1, CD11b, Gr1), BST-2, Siglec-H. Bar graphs quantify the expression of YFP in pDCs with mean ± SEM across two independent replicate experiments (*n* = 4–6 mice/group). p values were calculated with *p < 0.05; **p < 0.01; ***p < 0.001; ****p < 0.0001; ns, not significant, using two-tailed unpaired Student's *t*-test for flow-cytometry comparisons and Welch's *t*-test for comparison of IVM quantifications. Multiple linear regression analyses were applied for statistical analysis of the MSD plots. Data of WT *PTCRA*-EGFP depicted in (**B, C**) are the same as in *Figures 2B, C and 3B, C*; (**D**) are also in *Figures 2D and 3D*. WT *PTCRA*-EGFP/WT partial BM chimera in (**E**) are also in *Figures 2F and 3F*. These are included for ease of comparisons across conditions relative to WT.

The online version of this article includes the following video, source data, and figure supplement(s) for figure 5:

**Source data 1.** Raw values for *Figure 5A-F, H, I* graphs/data.

**Figure supplement 1.** Bone marrow cells from WT B6 mice were stained cell-surface expression of lin$^{neg}$ (CD3, CD19, NK1.1, CD11b, Gr1), BST-2, and Siglec-H and indicated chemokine receptors or no stain (fluorescence minus one, FMO).

**Figure supplement 1—source data 1.** Raw values for all graphs/data.

**Figure 5—video 1.** pDC movement and clustering in the bone marrow of naive compared to *Py*-infected *Sting1$^{Gt/Gt}$* mice.
https://elifesciences.org/articles/78873/figures#fig5video1

**Figure 5—video 2.** WT pDCs arrest and form large clusters in *Py*-infected *Sting1$^{Gt/Gt}$ PTCRA*-EGFP/WT chimeras.
https://elifesciences.org/articles/78873/figures#fig5video2

**Figure 5—video 3.** pDC forms large clusters in *Py*-infected *Sting1$^{Gt/Gt}$* compared to WT mice.
https://elifesciences.org/articles/78873/figures#fig5video3

---

CXCR3, CD86) underwent similar changes as in WT mice, suggesting either multistep activation or independent processes (*Figure 6—figure supplement 1A*). Interestingly, in these mice where CD169$^+$ MP lacked STING, the number of pDC in BM was ~1.5 times higher than in WT or *Sting11$^{Gt/WT}$* control groups (p value = 0.1, *Figure 6B*). Similar increases in numbers of pDC were also enumerated in the BM of whole-body STING-deficient (*Sting1$^{Gt/Gt}$*) or CD169$^+$ MP-depleted (DT-treated *siglec1$^{DTR/WT}$*) infected mice, consistent with retention of pDC in BM when CD169$^+$ MP or STING-dependent signals were absent (*Figure 5G–I* and *Figure 5—videos 2 and 3*). Collectively, these results show that STING signaling in CD169$^+$ MP during *Py* infection, provide pDC with key signal(s) to initiate type I IFN production and egress from the BM.

## STING-mediated activation of pDC is independent of CD169$^+$ MP loss

Upon *Py* infection, a majority of CD169$^+$ MP are rapidly lost (*Figure 1*). We hypothesized that STING signaling may control their loss and thus contribute to providing activating ligands and signals to pDC. We therefore assessed whether STING contributed to MP loss during *Py* infection. In whole-body *Sting1$^{Gt/Gt}$* mice, CD169$^{hi}$ but not F4/80$^{hi}$ MP, were partially protected from loss compared to WT control mice (by ~50%, *Figure 6C*), suggesting that this may prevent optimal pDC priming. However, in *siglec1$^{Cre/WT}$Sting1$^{Gt/F}$* mice where STING is selectively lacking in CD169$^+$ MP, both MP populations underwent similar loss as in WT mice, suggesting STING deficiency extrinsic to MP, protects CD169$^{hi}$ MP from loss. However, in *siglec1$^{Cre/WT}$Sting1$^{Gt/F}$* mice, pDC failed to initiate type I IFN transcription and accumulated in the BM compared to control groups (*Figures 5I and 6B*), consistent with a model in which STING signaling in CD169$^+$ MP provide activating signals to pDC that are unrelated to their loss. Of note, MP activation 36 hr post-infection, as measured by cell-surface modulation of MHC-II, CD80, ICAM-1, and Sca-1, was mostly comparable in all infected groups, likely ruling out a general activation defect of STING-deficient MP (*Figure 6—figure supplement 1B*). In summary, these data uncouple MP loss that occurs during *Py* infection from STING-mediated priming of pDC by CD169$^+$ MP.

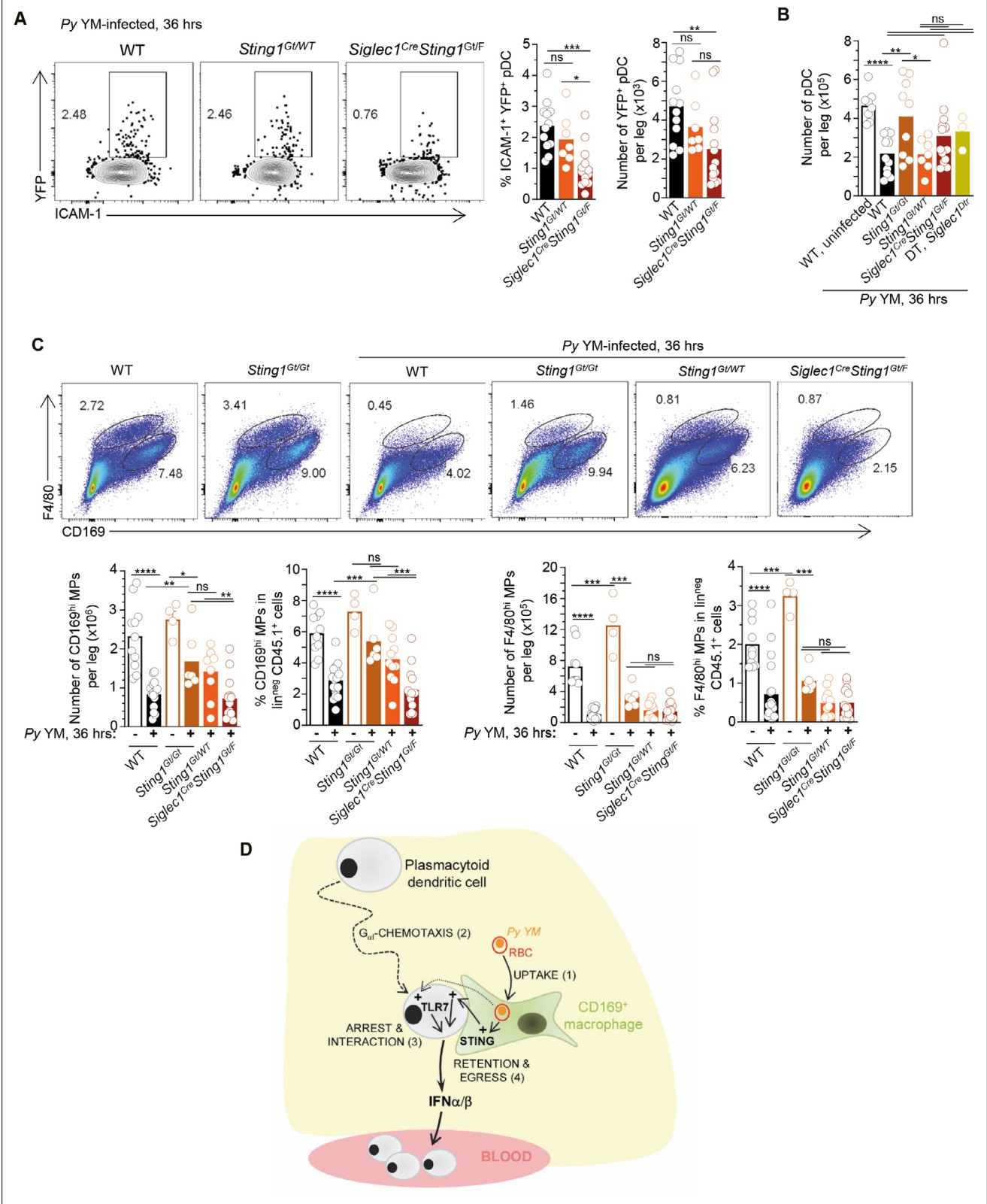

**Figure 6.** STING signaling in CD169+ MP is required for plasmacytoid dendritic cells (pDCs) production of type I interferon (IFN) and egress from the bone marrow (BM). WT-, $Sting1^{Gt/WT}$-, or $CD169^{Cre/Cre}Sting1^{Gt/F}$-$Ifnb^{YFP/YFP}$ reporter mice were infected with $5 \times 10^5$ $Py$-iRBCs. Thirty-six hours later BM cells were stained with live/dead and for cell-surface expression of $lin^{neg}$ (CD3, CD19, NK1.1, Gr1), CD11b, BST-2, Siglec-H, CD169, F4/80, and indicated cell-surface markers. (**A**) Representative FACS dot plots of YFP/ICAM-1+ pDC are shown, together with the proportion and numbers of YFP+ pDC/leg

*Figure 6 continued on next page*

Figure 6 continued

in summary bar graphs. (**B**) Number of pDC per leg in indicated mouse genotype. (**C**) Representative dot plots of CD169$^{hi}$ and F4/80$^{hi}$ MP in indicated mice. Bar graphs show numbers per leg and proportions. Data represent the pool of three independent replicate experiments with each symbol featuring one mouse ($n$ = 3–18 mice). p values were calculated with *p < 0.05; **p < 0.01; ***p < 0.001; ****p < 0.0001; ns, not significant, using two–tailed unpaired Student's $t$-test. (**D**) Working model: during *Py* YM malaria infection, CD169$^+$ MP rapidly take-up *Py*-infected RBC (step 1). Next, pDC, which is highly motile in the BM of naive mice, home to and establish interactions with CD169$^+$ MP at peak production of type I IFN by pDC (steps 2 and 3). This process is dependent on G$_{\alpha i}$-mediated chemotaxis. CD169$^+$ MP and TLR7 sensing in pDC are both required for pDC arrest. Lastly, STING signaling in CD169$^+$ MP is needed for pDC to initiate type I IFN secretion, regulate pDC clustering/retention, and egress from the BM to the blood (step 4).

The online version of this article includes the following source data and figure supplement(s) for figure 6:

**Source data 1.** Raw values for *Figure 6A-C* graphs/data.

**Figure supplement 1.** WT-, *Sting1*$^{Gt/WT}$-, or *CD169*$^{Cre/Cre}$*Sting1*$^{Gt/F}$-*Ifnb*$^{YFP/YFP}$ reporter mice were infected with 5 × 10$^5$ *Py*-iRBCs.

## Discussion

Here, we reveal a complex multi-cell choreography that orchestrates pDC production of type I IFN during malaria infection. We establish the existence of a functional interaction (direct or indirect) between CD169$^+$ MP and pDC in vivo that is needed to license pDC priming of type I IFN production in vivo during this infection. We link pDC arrest to their optimal priming. We show that CD169$^+$ MP take up parasite-infected RBCs in the BM. We find that both CD169$^+$ MP and pDC-intrinsic TLR7 sensing are required for pDC arrest, suggesting that MP is providing TLR7 ligands to pDC. We also show that MP-intrinsic STING sensing is required to activate pDC production of type I IFN, and enable their egress/release from the BM after priming (*Figure 6D*).

These findings are therefore consistent with a model in which CD169$^+$ MP in the BM act as first-line responders to infection, akin to other CD169$^+$ MP found in the LN and spleen (*Grabowska et al., 2018*). Depletion of these MP prior to infection prevents type I production by pDCs (*Spaulding et al., 2016*), indicating that these antigen presenting cells are non-redundant and uniquely capable of providing *Py* ligands and/or other innate signals to pDC in the BM. The fact that depletion of CD169$^+$ MP and cell-intrinsic lack of TLR7 prevent pDC arrest establishes that pDC motility, homing and functional interactions with MP are prerequisites to achieve type I IFN production. These results suggest that TLR7 ligands, which trigger pDC arrest and type I production, are also not freely diffusing in the BM and/or that in vivo, pDC require nucleic acids to be properly presented or packaged to access TLR7-containing endosomes in pDC. In either case, we propose that direct or indirect interactions with CD169$^+$ MPs are required to deliver *Py*-derived ligands to pDCs in the BM. This finding is consistent with prior studies showing pDC sensing of hepatitis C-infected hepatocytes via viral RNA-containing exosomes (*Dreux et al., 2012*; *Takahashi et al., 2010*) or cellular particles containing Dengue virus (*Décembre et al., 2014*). pDC was also reported to be directly infected by specific viruses such as herpes simplex virus and arenaviruses, and to utilize specialized intracellular processes like autophagy to sense viral nucleic acids and initiate type I IFN production (*Lee et al., 2007*; *Macal et al., 2012*; *Assil et al., 2019*). Given the very low frequency and numbers of pDC but their rapid response, it is likely that pDC are largely using such indirect but more effective modes of recognition. The formation of specialized cell–cell structures between pDC and infected cells or cells presenting pathogen-derived molecules, is indeed proposed to mediate both uptake and sensing of TLR ligands and the local delivery of type I IFN in interferogenic synapses (*Assil et al., 2019*).

Proper clustering and arrest of pDCs is a TLR7 guided process. TLR7 is required in pDC for their arrest, and type I production, as well as extensive remodeling of pDC expression of chemokine receptors and adhesion molecules, likely promoting stable contacts with activated, *Py*-loaded MPs. In vivo, pDC are recruited via CXCR3- and CCR5-dependent chemotaxis to virus-infected subcapsular draining LN MP or CD8$^+$ T cells, respectively, for local delivery of type I IFN, rapidly restricting early virus spreading while promoting effective T cell priming (*Brewitz et al., 2017*). Given that pDC expresses multiple chemotactic receptors, and some are significantly up- or downregulated during *Py* infection (e.g., CCR9, CXCR3, CX3CR1), these processes are likely to be redundant. Consistent with this interpretation, blockade of CXCR3-mediated chemotaxis failed to recapitulate all G$_{\alpha i}$-mediated chemotaxis blockade. Even co-blockade of several adhesion molecules (LFA-1, VLA-4) involved in classical leukocyte adhesion and clustering, did not abrogate the production of type I IFN by pDC,

suggesting that other chemotactic and adhesion mechanisms are involved and further investigations are needed.

Finally, we establish TLR7-dependent arrest and interaction of pDC with CD169+ MP. We also show that full activation of pDC requires STING signaling in CD169+ MP and that MP control pDC arrest. It was therefore unexpected that we did not find STING controls pDC arrest. However, we revealed that in STING-deficient mice, pDC formed significantly greater size clusters and accumulated in higher numbers in the BM compared to WT counterparts. This result was confirmed in mice lacking STING in the radioresistant compartment and only in CD169+ MP, consistent with a model where STING-dependent signals control pDC egress from the BM, and as a consequence, the organization of pDC clusters. In our previous work (*Spaulding et al., 2016*), we reported that pDC numbers in the BM of *Py*-infected mice decreased after peak type I IFN production while that of blood pDC increased. Larger clusters and increased numbers of pDC in STING-deficient mice, and in mice with STING-deficient CD169+ MP, may indicate that pDC are not fully activating and thus remain docked, awaiting for additional signals. These large clusters of pDC are further evidence that pDC priming is spatially segregated in the BM and that activation signals are not freely diffusing. These results connect the choreography of the cell–cell signal transmissions with pDC changes in arrest and migration for efficient regulation. Other signals that need to be elucidated are likely to regulate these cellular interactions in vivo.

In particular, which STING-dependent signal(s) are delivered from *Py*-activated CD169+ MP to pDC remains unknown. STING was originally identified as a cytosolic nucleic-acid sensor, which triggering leads to the induction of a robust type I IFN response via IRF3 and NFkB, and a rapid antiviral state that enables the control of viral infections (*Zhong et al., 2008*; *Ishikawa and Barber, 2008*). While pDC can secrete type I IFN independently of type I IFN signaling during VSV and MCMV infections (*Tomasello et al., 2018*; *Barchet et al., 2002*), pDC are also known to require type I IFN autocrine signaling when responding to TLR7/9 synthetic ligands in vivo (*Asselin-Paturel et al., 2005*). Thus, type I IFN may be the STING-dependent missing signal given by CD169+ MP during malaria infection. Several studies provide evidence that CD169+ subcapsular marginal zone and sinus MP in the spleen and LN, respectively, may produce type I IFN in response to viral infections (VSV, Vaccinia, MCMV) (*Gui et al., 2019*; *Brault et al., 2018*; *Gaidt et al., 2017*). However, since STING also triggers the proinflammatory transcriptional regulator NFkB, other cytokines could be the missing signal. STING signaling can also initiate distinct inflammatory cell deaths and autophagy as a defense mechanism. Among STING-mediated cell death, lysosomal cell death, pyroptosis, necroptosis, and NETosis were described to occur in myeloid cells in response to cGAS-STING sensing/signaling of viral DNA (*Gui et al., 2019*; *Brault et al., 2018*; *Gaidt et al., 2017*; *Lood et al., 2016*). While we observed a major loss of CD169+ MP in the BM (>60%) of *Py*-infected WT mice when type I IFN levels peak (30–36 hr), STING-deficient MP underwent similar loss, most likely excluding STING involvement in MP death. However, STING-driven autophagy could provide live CD169+ MP with *Py*-derived nucleic acids and other products for optimal presentation to pDC.

STING is generally shown to require cGAS for cytosolic double stranded DNA recognition (*Sharma et al., 2011*) and the production of STING-activating cyclic dinucleotides second messengers (*Chen et al., 2016*). Our results are discrepant with a prior study that also used lethal murine *Py* YM, and concluded that STING sensing of *Py* parasites in pDC prevented strong type I IFN responses by pDC through pDC-intrinsic SOCS1-mediated inhibition of TLR7/MyD88 signaling (*Yu et al., 2016*). In our previous work, as well as that of other groups (*Spaulding et al., 2016*; *Sharma et al., 2011*), blocking type I IFN production and signaling, increased mouse resistance to severe *Py* YM and *P. berghei* and non-lethal *P. chabaudi* and *Py* 17XNL murine malaria, respectively. We neither noted increased type I IFN levels in *Sting1*$^{Gt/Gt}$ mice, nor increased IFNβ production by STING-deficient pDC, and *Sting1*$^{Gt/Gt}$ mice did not exhibit greater resistance to *Py* YM either. Also, while the STING pathway may be active in pDCs, current experimental evidence is scarce (*Reizis, 2019*; *Bruni et al., 2015*), consistent with our finding that STING sensing occurs in the CD169+ MP, extrinsically to pDCs. We favor the interpretation that the *Py* YM strain used in this previous work, is likely to exhibit genetic differences with the repository *Py* YM strain MRA-755 we have used, which may account for the different outcomes. However, taken together, our results that type I IFN signaling negatively impact malarial infection outcomes, appear to be more consistent with the general consensus across several murine models of infection and laboratories.

In conclusion, our analysis of the dynamic behavior of pDC in vivo by IVM directly in the BM of mice undergoing severe murine malaria, reinforce a model in which pDC sensing of their ligands in vivo is a highly coordinated and orchestrated process. We establish that CD169$^+$ MPs, a subset of tissue-resident MP that act as essential sentinels of the immune system constantly sampling the lymph and blood (*Sung et al., 2012*; *Kastenmüller et al., 2012*; *Perez et al., 2017*; *Miyake et al., 2007*), play a key role in regulating pDC activation in vivo. While we establish the existence of a functional inter-action between these MP and the pDC in the BM of malaria-infected mice, further investigations will be needed to determine (1) the nature of the STING-dependent signals derived from CD169$^+$ MP that allow pDC to achieve type I IFN-producing capacity and egress the BM to the blood and (2) whether this interaction is direct or not, and may also take place in other tissues and infections or autoimmune diseases.

# Materials and methods

## Mice

C57BL/6J wild-type (WT B6), $Tlr7^{-/y}$ (*Lund et al., 2004*) (strain 8380, Jackson Labs), $Ifnb^{mob/mob}$/$Ifnb\text{-}yfp^{+/+}$ (denoted $Ifnb^{YFP/YFP}$) (*Gaya et al., 2015*) (strain 10818, Jackson Labs), $Sting1^{Gt/Gt}$ (*Sauer et al., 2011*) (strain 17537, Jackson Labs), $Sting1^{F/F}$ (strain 031670, Jackson Labs), $Rosa26^{LoxP\text{-}STOP\text{-}LoxP(LSL)\text{-}tTomato}$ (strain 007914, Jackson Labs), and $Rosa26^{LSL\text{-}YFP}$ (strain 006148, Jackson Labs) mice have been described and were purchased from the Jackson laboratory. *PTCRA*-EGFP mice (129 background) (*Shigematsu et al., 2004*) and were obtained from Dr. Boris Reizis (NYU, New York, NY) and back-crossed to WT B6 mice for 10 generations. $Tlr7^{-/y}$ and $Sting1^{Gt/Gt}$ *PTCRA*-GFP as well as $Tlr7^{-/y}$ and $Sting1^{Gt/Gt}$, $Ifnb^{YFP/YFP}$ reporter mice were obtained by intercross, respectively, for use in IVM and FACS assays. $siglec1^{DTR/DTR}$ and $siglec1^{Cre/Cre}$ mice, a kind gift from Dr. Masato Tanaka (Riken, Japan) (*Miyake et al., 2007*) expressed the human diphtheria toxin receptor (DTR) or the Cre recombinase knocked-in to the exon 1 of the CD169-encoding gene the mice. Mice were used as heterozygous for experiments ($siglec1^{DTR/WT}$, $siglec1^{Cre/WT}Sting1^{Gt/F}Ifnb^{YFP/YFP}$, $Sting1^{Gt/F}Ifnb^{YFP/YFP}$, $siglec1^{Cre/WT}Rosa26^{LSL\text{-}tTomato/WT}$, $siglec1^{Cre/WT}Rosa26^{LSL\text{-}YFP/WT}$), obtained by intercrossing of the relevant strains above. All mice were housed and bred in specific pathogen-free conditions at the Albert Einstein College of Medicine. Eight- to twelve-week-old and sex-matched males and female mice were used for all experiments.

Partial BM chimeras $siglec1^{DTR/WT}$ or WT B6 recipient mice were exposed to 5 Gy total body irradiation (500 rads) and reconstituted with $2 \times 10^6$ BM cells isolated from either $Tlr7^{-/y}$ or WT *PTCRA*-EGFP reporter mice. Chimerism of reconstituted mice was checked ~6 weeks later in the blood, before infections were conducted. Blood chimerism of donor-derived HSC was ~30%.

## *Plasmodium* infections

*Plasmodium yoelii yoelii* (17X), clone YM parasites (stock MRA-755) was obtained from BEI Resources. *P. yoelii* YM-infected red blood cells (iRBCs) from a frozen stock (stored in liquid nitrogen in Alsever's solution and 10% glycerol) were injected intraperitoneally (i.p.) into a eight- to twelve-week-old WT B6 mouse, and grown for ~4 days. When parasitemia reached 2–10%, $5 \times 10^5$ . *yoelii* YM iRBCs were inoculated intravenously (i.v.) in a volume of 200 µl PBS into each experimental mouse.

## Preparation of cell suspensions for flow cytometry (FACS) analysis

Blood was collected in K2 EDTA(Ethylenediaminetetraacetic) tubes and red blood cells lysed with NH$_4$Cl buffer (0.83%, vol/vol). BM was harvested from the femur and tibia by removing epiphyses to expose marrow cavity and placing into a 0.5-ml Eppendorf with a punched hole in the bottom made with an 18.5 G needle, nested into a 1.5-ml Eppendorf tube. Tubes were next centrifuged at 10,000 × *g*, 4°C, and cells resuspended in media for further used for further analyses (see below).

## Cell staining for FACS analysis

Cell suspensions were stained with LIVE/DEAD Fixable Aqua (Thermo Fisher Scientific) or Ghost Dye Red 780 (Tonbo) in PBS for 30 min at 4°C, then incubated with Fc-block CD16/32 (clone 2.4G2) for 20 min at 4°C. Cell suspensions were next incubated in Ab mixes in FACS buffer (2 mM EDTA, 2% fetal bovine serum [FBS], 0.02% sodium azide in PBS) for 20 min at 4°C. For cell counts, we utilized 50,000 counting beads (SpheroTech) added to cells isolated from one leg or one spleen. Stained cells

were collected either on a FACS BD LSR-II or Aria III. A list of all mAbs and sera used in the study with full information is provided in *Supplementary file 1*. Data were analyzed using FlowJo version 9.6.2 (TriStar).

## RBC labeling

Red blood cells were harvested from uninfected or *Py* YM-infected WT B6 mice with ~20% parasitemia by submandibular puncture into K2 EDTA tubes and washed twice with 10 ml 1× $Mg^{2+}Ca^{2+}$-free PBS. Peripheral blood was pelleted by centrifugation at $800 \times g$ and 4°C for 5 min. RBCs were gently resuspended up to a volume of 5 ml in 1× $Mg^{2+}Ca^{2+}$-free PBS, and overlaid on top of 5 ml room temperature Ficoll-Plus in a 15 ml Falcon tube which was centrifuged at $1000 \times g$ for 15 min at 20°C without breaks. RBC pellet was washed in 5 ml $Mg^{2+}Ca^{2+}$-free PBS, before staining with 5 µM 5-carboxytetramethylrhodamine (TAMRA-Red, ThermoFisher Scientific, Catalog#C6121) in PBS 2% FBS for 20 min at 37°C, 5% $CO_2$ and in the dark. Labeled RBCs were then washed in 10 ml PBS, and 50 µl of packed RBCs recovered in 200 µl PBS were injected i.v. by retro-orbital route into recipient mice.

## PT and in vivo mAb treatments

PT (Sigma Millipore #516560) was reconstituted in 100 mM sodium phosphate, 500 mM sodium chloride, pH 7.0 and stored at −80°C. Mice were treated with 2.5 µg/µl PT in 100 µl of 1× PBS, i.v. at the time of infection. For mAb injections, mice received 250 µg of each mAb as specified in the figure panel, i.v. 30 min prior infection (*Supplementary file 1*).

## Two-photon preparation, imaging, and analysis

In vivo imaging of BM was conducted as described previously (*Pitt et al., 2015*). *PTCRA*-EGFP reporter mice were anesthetized with isoflurane (2.5–3% for induction, 1–2% for surgery/imaging) admixed in 1:1 $O_2$: air mixture at a flow rate of 1 l/min. Once induction was achieved, a gas mask was secured over the nose and stabilized using Velcro on a stage maintained at 37°C. The mouse anesthetic plane was ensured through the lack of response to toe and/or tail pinches. Level of isoflurane was adjusted accordingly to ensure the mouse maintained appropriate sedation with a stead, non-labored respiratory rate between 60 and 80 breath/min. Leg hair was then shaved using a surgical blade and a small amount of soap and water to facilitate hair removal. Loose hair and residual soap/water were removed with a kimwipe and the leg dried. Subsequently, the mouse torso was secured on the stage with Velcro under the diaphragm so as to not interfere with breathing, and the leg stabilized with a piece of scotch tape with the mouse is a supine position and the foot everted to reveal the medial surface of the leg. Then, the medial aspect of the tibia was surgically exposed by making a small incision at the level of the medial malleolus, avoiding the saphenous artery, to expose the tibial bone. A window was secured to the leg using vet bond, sliding a small tooth of the window under the tibial bone and then fixed to the stage using screws. Then, the exposed bone was shaved using a microdrill to a 100-µm thickness. Once optimal thinness was achieved, a ring of vacuum grease was drawn around the imaging area, then filled with a drop of 37°C lactated ringers (Thermo Scientific Cat#R08432), and the mouse transferred to the microscope chamber maintained at 37°C. Images were collected on an Olympus FV-1000MPE upright laser scanning microscope using a MaiTai DeepSee Laser (Spectraphysics) set to a wavelength of 910–920 nm. BM was visualized using a ×25 NA 1.05 objective and magnified 1.5–2×. Time lapse stacks with 5 µm slices spanning a total depth of 50 µm were captured at an acquisition of 4.0 µs/pixel using. To visualize CD169[+] MP, mice were intravenously injected with 5 µg of anti-CD169 (SER4) antibody (BD Pharmingen) conjugated to PE (or FITC for *Py*-iRBC uptake experiments), 16 hr prior to imaging (*Gaya et al., 2015*).

Movie analysis: 4D image sets were analyzed on Volocity 6.3 (Improvision). Tracking was conducted using semi-automated scripts using spot function. For pDC identification, Channel 3 subtracted from Channel 2 using imaging processing -> channel arithmatics function. XY diameter set to 10 µm, threshold 7 for brightness, max distance 8 µm, max gap 0. Filtered to include only pDCs tracked >300 s. For cell-contact duration between pDC and CD169[+] MP, analysis was conducted using Kiss and Run Analysis built-in plugin extension between pDC spots and MP surfaces (Imaris, Matthew Gastinger). Distance transformation function used with threshold set to 5 µm for distance. Data obtained from contact durations were tallied up for all pDC in the field in contact with CD169[+] MP. Data were

pooled from three or more mice from independent experiments. Supplementary movies were assembled using After Effects (Adobe). Contrast and saturation were enhanced to allow for greater ease of pDC visualization above MP for inclusion in publication (post-analysis).

Two-dimensional plots were created at a higher resolution to our specification using Matlab. Instructions and script hyperlinked: ./XYZ plots/Read Me.docx; ./XYZ plots/XY_Plots_ZB_10082019.m.

## Statistics

All statistical tests were run with GraphPad Prism 8. Significance is depicted as follows: $*p < 0.05$; $**p < 0.01$; $***p < 0.001$; $****p < 0.0001$. All graphs show mean ± standard error of the mean. The unpaired *t*-test was used for comparing two groups related to FACS data. Welch's *t*-test was used for comparing two groups related to IVM analyses. Multiple linear regression analyses were applied to test if the factor conditions (time and group) predicted displacement for statistical analysis of the MSD plots, assuming a gaussian distribution of residual and least squares.

## Acknowledgements

We thank the AECOM FACS core facilities. We thank Ilseyar Akhmetzyanova for helping with initial training on intravital microscopy and Zachary Benet for providing the MATLAB script to make the displacement charts featured in this manuscript.

This work was funded by the National Institutes of Health Grants (NIH/NIAID) AI103666 and Hirschl Caulier Award to GL and AI128735 to DF and GL. JM was supported by NIH MSTP training grant T32 GM7288-46 and F31 HL147470.

## Additional information

### Funding

| Funder | Grant reference number | Author |
|---|---|---|
| NIAID | AI103666 | David Fooksman Gregoire Lauvau |
| NIH | T32 GM7288 | Jamie Moore-Fried |
| NIH | F31 HL147470 | Jamie Moore-Fried |

The funders had no role in study design, data collection, and interpretation, or the decision to submit the work for publication.

### Author contributions

Jamie Moore-Fried, Conceptualization, Data curation, Formal analysis, Investigation, Visualization, Methodology, Writing - original draft, Project administration, Writing - review and editing; Mahinder Paul, Formal analysis, Investigation, Methodology, Project administration, Writing - review and editing; Zhixin Jing, Software, Formal analysis, Supervision, Validation, Methodology, Writing - review and editing; David Fooksman, Conceptualization, Resources, Software, Formal analysis, Supervision, Funding acquisition, Validation, Investigation, Visualization, Methodology, Project administration, Writing - review and editing; Gregoire Lauvau, Conceptualization, Resources, Formal analysis, Supervision, Funding acquisition, Validation, Investigation, Visualization, Methodology, Writing - original draft, Project administration, Writing - review and editing

### Author ORCIDs

Gregoire Lauvau http://orcid.org/0000-0002-3050-2664

### Ethics

This study was carried out in strict accordance with the recommendations by the animal use committee at the Albert Einstein College of Medicine under protocol number #20171113 and 00001375. The institution is accredited by the 'American Association for the Use of Laboratory Animals' (DHEW Publication No. (NIH) 78-23, Revised 1978), and accepts as mandatory the NIH 'Principles for the Use

of Animals'. All efforts were made to minimize suffering and provide humane treatment to the animals included in the study.

### Decision letter and Author response
Decision letter https://doi.org/10.7554/eLife.78873.sa1
Author response https://doi.org/10.7554/eLife.78873.sa2

---

## Additional files

### Supplementary files
• Supplementary file 1. Table of antibodies used for FACS, microscopy, and in vivo depletion experiments.

• MDAR checklist

### Data availability
All data generated or analyzed during this study are included in the manuscript and supporting file; Source Data files have been provided for all figures. There is no restriction of access.

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
