## [Editor Report]

This study presents valuable mechanistic findings demonstrating the requirement of CD169+ macrophage intrinsic STING signaling in regulating plasmacytoid DC motility/arrest and their activation. The evidence supporting the claims of the authors is solid, and their data suggest that adequate type I IFN production by pDC may also require macrophage intrinsic STING signaling. The work will be of great interest to the immunology community.

---

## [Decision Letter]

**Decision letter after peer review:**

[Editors’ note: the authors submitted for reconsideration following the decision after peer review. What follows is the decision letter after the first round of review.]

Thank you for submitting your work entitled "Spatio-temporal organization of plasmacytoid dendritic cell priming by CD169+ macrophages in malaria-infected mice" for consideration by *eLife*. Your article has been reviewed by 3 peer reviewers, and the evaluation has been overseen by a Reviewing Editor and a Senior Editor. The following individuals involved in review of your submission have agreed to reveal their identity: Michael L Dustin (Reviewer #3).

Our decision has been reached after consultation between the reviewers. Based on these discussions and the individual reviews below, we regret to inform you that your work will not be considered further for publication in *eLife*.

The reviewers commend you on your efforts to extend your previous work and design a robust methodology to assess selective interactions of parasitized RBCs with macrophages. However the reviewers also noted major concerns with conflicting data and overstated results that significantly detracted from their enthusiasm. Significant concerns were expressed regarding the demonstration of interactions between CD169+ macrophages and pdcs, which is a major conclusion of the paper. Furthermore, two reviewers were concerned that there was not a clearly defined delineation between this work and your previous work. Specific comments are listed below.

*Reviewer #1:*

In this study, the authors extend previous work published from their lab showing that CD169+ macrophages license plasmacytoid dendritic cells (pDCs) to make type I interferon in the bone marrow during P. yoelii infection. Though there are some interesting points made in the manuscript, overall the manuscript reads more as a compilation of findings following up on their previous PLOS Pathogens paper, rather than a coherent story. Furthermore, some of the more novel conclusions in the manuscript are not well supported by the data.

1. In several parts of the manuscript, it is difficult to distinguish by the writing what was previously published by this group and what is new to this manuscript.

2. From Figure 1, the authors conclude that CD169 is involved in the binding and uptake of infected RBC. However, the authors mis-state what the data show. They state "we found that CD169 het mice, BM MPs had significantly higher frequencies of both extra and intracellular TAMRA-Red+…" However, Figure 1D shows a lower frequency of extracellular iRBCs, but no difference in intracellular RBCs. Though the "representative" flow is picked to show a >5x difference in extracellular RBCs, the number of both intra and extracellular RBCs bound/internalized by macrophages is significantly different, showing that the number of BM MPs is what is changing. In fact, the authors show the F4/80hi cells go down in number after infection in Figure 2, consistent with this interpretation.

3. Much of the manuscript uses CD169-/- mice and the authors state that the CD169hi macrophages still exist in these mice, however it is critical to show this to be able to interpret the data using these mice as the authors have already shown that these cells are critical for pDC clustering and IFNa production.

4. In Figure 2, there are two contradictory pieces of data. In Figure 2A, CD11a is show to be downregulated in infected mice, while in Figure 2D, CD11a is unchanged compared to uninfected mice.

5. Although the authors acknowledge that PT treatment has indirect effects, they still conclude that pDC motility is required for pDC IFNa production and fail to address the possibility that motility of other cell types, including CD169+ macrophages, being involved.

6. In Figure 7, the authors demonstrate that STING is required independently of cGAS for efficient type I interferon production by pDC. However, the authors had already demonstrated a role for STING extrinsic to pDC in their previous publication. Therefore, the only new piece of data here is that it is independent of cGAS, which is interesting, but needs to be followed up on in its own right.

7. This manuscript would be greatly enhanced by a model of what had been shown previously and what is shown in this manuscript.

8. Use of statistics throughout the manuscript is not consistent and tests used are not noted in figure legends. Sometimes only 2 groups in a figure are compared and sometimes all groups, even within one figure panel, with no justification.

*Reviewer #2:*

Moore et al. characterize the phenotype and movement of pDCs in the bone marrow of infected mice, and test several pathways (TLR7, STING, cGAS, GPCRs) for their role in regulating pDC motility, phenotype, and interferon production. There are several nice experiments here. Unfortunately, the authors grossly misrepresent or overinterpret their findings and extrapolate to a model of cell-cell interaction for which they provide little evidence. This is clearly a follow-up study to their 2016 PLoS Pathogens paper, but I'm not convinced it stands on its own; at a minimum it needs extensive textual revision to accurately represent the data, and once it has done that, I'm not sure it meets broad enough interest for an *eLife* paper.

My main concern is that the text misrepresents what the data actually show. E.g., the title and abstract claim that CD169+ macrophages coordinate the localization and function (IFN secretion) of pDCs in a STING- and TLR7-dependent manner, but there is not a single figure in the paper that quantifies interactions between macs and pDCs. (The handful of microscopy images are not sufficient to make conclusions about these interactions-and the authors do not even attempt to do this, e.g. by using arrowheads to mark interacting cell clusters). The authors do state (p. 9) that pDC did not cluster around CD169+ macs in infected TLR7 KO mice compared to WT mice, but they show no data, and cite their previous paper for the WT observation. Further, there are no data to show that STING and TLR7 are acting within macs, rather than pDC (or another cell type), to affect arrest or clustering of pDC. I gather that the authors may have shown in their 2016 paper a requirement for mac-expressed STING in pDC production of interferon, but that doesn't mean the same is true for pDC motility/ clustering. (And it doesn't tell us anything about the TLR7 requirement.) The authors establish in Figure 1 that macs take up infected RBCs, but they do not show that pDCs do NOT take up iRBCs (which could perhaps demonstrate indirectly that pDCs must get their activating signals from macs). There is precedent for direct pDC uptake of malaria parasites (Wykes et al. PNAS 2011), though controversial… still, in this paper at least, the authors have not demonstrated that any interactions with CD169+ macs are controlled by STING/ TLR7/GPCR, or required for IFN production by pDCs.

I also think the involvement of TLR7 in pDC arrest-a key point of the paper-is overstated. The section heading (p.8) states that "pDC arrest in Py-infected bone marrow is TLR7-dependent." But as the authors note, TLR7 KO pDCs have much higher baseline activity. They measure motility in four ways: track velocity, displacement rate, MSD, and slope of MSD. The latter two clearly show that TLR7 KO pDC still arrest after infection; the stats are not there to show whether the magnitude of their arrest is significantly different from WT. One measurement of motility (velocity) shows that TLR7 KO cells do differ from WT in that they don't slow down after infection. The final measure, displacement rate (3C- bottom) is not convincing: WT mice decrease displacement rate significantly after infection and TLR7 KO mice don't, but the WT infected group has an n of 7, and the TLR7 group only has 3 mice and is trending downward. It seems very possible that with more mice, this would become significant. So. at least two, and possibly 3 of the 4 measures of motility show NO difference between WT and TLR7-/- pDC. Given this, I think the finding is way overstated. The authors can still say that TLR7 plays some role in pDC motility, but they need to qualify this quite a bit (which they do on p. 9, but not throughout much of the paper, including the abstract).

A third unsupported conclusion is that "PT treatment led to immobilization of pDC, and their lack of visible clustering with MP" (p. 10, Figure 5A, B). The authors show that pDCs are largely nonmotile in infected, PT-treated mice, but they do NOT compare to untreated controls, in which we know that pDCs are arresting and largely nonmotile (Figure 3A second panel). Is there a difference between motility in infected, PT-treated versus untreated mice? This is critical to show. Further, as mentioned before, there is no measurement of clustering with macs, so the authors should not assert that PT treatment alters this. Even the assertion that pDC are "morphologically distinct" after PT treatment seems overstated (p. 10); Figure 5C shows no difference in shape factor and a trending but non-significant difference in longest axis.

*Reviewer #3:*

pDC are important cells in the innate immune system that produce type I IFN (IFNab) in response to infection or injury. This group published in 2016 that pDC are highly motile in the bone marrow, but arrest in proximity to macrophages in the context of experimental malaria. Here, they further investigate the role of the CD169 receptor, TLR7, STING and cGAS innate sensors and GPCR motility receptors on this response. The work more clearly defines the role of the innate sensing pathways and chemokine receptors in the motility phenotype and the functional endpoint of IFNab production. There are a number of very nice aspects to this study. They have developed a robust method to detect the selective interaction of parasitized RBC with macrophages and to assess whether or not they are internalized on captures on the surface of macrophages. The baseline motility is completely pertussis toxin sensitive, although it was not possible to link the deceleration to an IFNab responsive chemokine receptor/chemokine pair. The interesting result is that the control of motility is necessary for the response, but not sufficient for IFNab production. A weakness of the paper is that the genetic manipulations effect both the macrophages and the pDC, so it's not possible to determine, for example, if TLR7 or STING are mainly operating in the pDC or the macrophage, or both.

The number of experiments is not quite enough to nail down the pDC migration phenotype of the TLR7 deficient mice. Three of the 4 mice analysed have more persistent migration such that the MSD is clearly greater. While the trend is clear the statistical significance is marginal, but this seems to be due to a limited samples size. So a couple of more experiments would be helpful. The data on the STING KO confirm that they can nail this when the migration phenotype is similar between two mouse strains.

[Editors’ note: further revisions were suggested prior to acceptance, as described below.]

Thank you for resubmitting your work entitled "CD169^+^ macrophages orchestrate plasmacytoid dendritic cell arrest and retention for optimal priming in the bone marrow of malaria-infected mice" for further consideration by *eLife*. Your revised article has been evaluated by Tadatsugu Taniguchi (Senior Editor) and a Reviewing Editor.

The manuscript has been improved but there are some remaining issues that need to be addressed, as outlined below:

Essential revisions:

1) CD169+ Macs a) capture infected RBC and b) interact with PDC.

2) Show FN Type 1 secretion.

The strategies proposed by reviewers could include:

Using breeding of PTCRA-eGFP mice and CD169Cre x Rosa fluorescent reporters.

Use of CD169-Knockin reporter.

Use of CD169 Cre x Rosa stop flox-reporter to image CD169+ Macs and perform shield chimera and graft the BM of PTCRA-eGFP mice into CD169 Cre x Ai14 mice for example.

*Reviewer #1:*

This is a revised manuscript by a group that has published on this topic before in 2016 (Plos Pathogen paper). They largely extend their previous findings and add some mechanistic studies demonstrating the requirement of CD169+ macrophage intrinsic STING signaling in regulating pDC motility/arrest. Furthermore their data suggest that adequate type I IFN production by pDC may also require macrophage intrinsic STING signaling.

Since I was not part of the initial review I evaluated the previous reviewer comments and assessed how well the authors responded to the criticisms. I agreed with the majority of the concerns raised by the previous reviewers. Furthermore, the authors have attempted to strengthen the study in good faith and have included new studies. In particular, in inclusion of experiments where CD169 conditional STING-KO mice are used adds to the novelty of this study.

The study is largely an extension of their previous work and in the revised manuscript the results for the most part support their conclusions. The use of dynamic imaging to assess pDC dynamics in the bone marrow is interesting and will add new avenues for further studies.

This study also adds to our knowledge of how pDCs get activated and produce type I IFNs in vivo. The mechanisms of how pDCs get activated in vivo still remain poorly understood, however, this report begins to shed some light.

Some deficiencies still remain, which will be discussed below:

Concerns that still remain:

1. Can the authors comment on the concern that use of anti-CD169 antibody for imaging the macrophages in vivo may prevent adequate uptake of infected RBCs or parasites in vivo?

2. Could the authors comment more on the physiological significance of type I IFN during malaria infection. Are pDCs needed for clearance of the infection and if type I IFN is required at all for the benefit of the host. This is a rather controversial topic and a clearer discussion taking their own studies in to account is warranted.

3. The fact that the motility of pDCs in TLR7 KO mice even under a steady state is so much higher still makes the interpretation of the data in Figure 3 difficult.

4. The authors make the claim that CD169+ macrophage intrinsic STING signaling is required for the production of Type I IFN by pDCs. However, they only use the reporter mouse for these studies, and the imaging is not very clear. Thus, the authors should show ELISA data in the serum and bone marrow to complement their imaging studies.

5. The CD169+ macrophages themselves can produce copious amounts of Type I IFNs. Can the authors eliminate the possibility of paracrine IFN secreted by the macrophages as another mechanism by which pDCs respond to and produce more IFN – especially in studies where CD169-DTR mice are used or in STING conditional KO studies? Here too, assessing IFN production will be valuable.

6. Finally, the authors should be careful about making any substantive conclusions with respect to pDC – CD169+ macrophage interactions. In this respect the dynamic imaging is not as convincing. The authors themselves admit that due to the high density of macrophages in the BM "it makes it difficult to establish a functional interaction"

*Reviewer #2:*

This revised version investigates the role of CD169 BM macrophages in the activation of pDC following the injection of P.yoelii infected RBCs. It aims at demonstrating that BM CD169+ macrophages are key regulators of pDC retention and activation, in a TLR7 and STING dependant manner.

Overall, the take home message appears valid but some experiments need to be performed in order to exclude technical artifacts:

1. TAMRA and PE appear to have very similar excitation and emission spectra. In addition, they both emit in the "Red" channel, which is typically highly autofluorescent. Why did the authors chose these two dyes to specifically label RBC and CD169+ Macs? The use of green/red (or blue/red or blue/green) fluorochromes might have improved the quality of the movies.

2. Reviewer 1 is correct when he/she states that PT treatment has indirect effects than can alter pDC motility and IFNa production. As the authors fail to identify the chemokines (or PT sensitive signalling pathways) that are at play, it is incorrect to state that pDC motility is required for pDC IFNa production. The authors should make this very clear in the manuscript.

3. Figure 1 does not convince me that CD169 Macs pick up infected RBC. It is well documented that CD169+ macs are fragile (at least in lymphoid organs) and that most CD169+ cells analyzed by flow are not even macrophages (PMID: 22675532 and more recently PMID: 34818538). As IVM lacks resolution, the authors should perform the same experiment in a PTCRA-eGFP mouse and stain fixed bone sections for CD169/GFP expression and then quantify the amount of fluorescent RBC captured by pDC and CD169+ macs. In addition, as the authors possess CD169Cre mice, I am wondering why they did not cross this strain to a fluorescent reporter such as the Rosa-RFP line to (a) visualize CD169+ macs by IVM and (b) prove that pDC are not fate-mapped in this model, and thus not targeted in CD169DTR mice.

4. Related to the previous points, I am concerned by FigS2C and sup movie 3. Both are showing data obtained in CD169 DTR mice treated with DT. While flow cytometry showed a 6 fold reduction of CD169+ macs upon DT injection, IVM does not show a similar decrease. If true, it suggests that one technique is not reliable to observe CD169+ macs. Can the authors comment on this?

5. In their reply to the reviewer's comments, the authors seem to have addressed the other concerns of the referees, including the use of a CD169Cre sting FF model to ablate STING in CD169+ Macs.

---

## [Author Response]

[Editors’ note: the authors resubmitted a revised version of the paper for consideration. What follows is the authors’ response to the first round of review.]

Reviewer #1:In this study, the authors extend previous work published from their lab showing that CD169+ macrophages license plasmacytoid dendritic cells (pDCs) to make type I interferon in the bone marrow during P. yoelii infection. Though there are some interesting points made in the manuscript, overall the manuscript reads more as a compilation of findings following up on their previous PLOS Pathogens paper, rather than a coherent story. Furthermore, some of the more novel conclusions in the manuscript are not well supported by the data.1. In several parts of the manuscript, it is difficult to distinguish by the writing what was previously published by this group and what is new to this manuscript.

The revised version now clarifies what was published before and what is new. Briefly, our prior study published in PLoS Pathogens in 2016, established that type I IFN receptor deficient mice resist lethal *P. yoelii (Py)* infection and that pDC are the major early producers of type I IFN in response to this infection. This work also showed that type I IFN production is initiated in the BM of infected mice via pDC-intrinsic MyD88/TLR7 signaling and upon priming by CD169^+^ BM MP. It further established the requirement for STING sensing in radioresistant cells, presumably the CD169^+^ MP. Lastly, we also provided in this work the first evidence, using intravital mutiphoton imaging (IVM) of the BM of living mice, that pDC arrested next to CD169^+^ MP at their peak production of type I IFN (36 hrs post *Py* infection).

In the current work, we investigated if CD169^+^ MP and parasite sensing via TLR7 and STING regulates pDC dynamic behavior and their interactions with CD169^+^ BM MP in vivo. Since CD169^+^ MP are required to prime pDC production of type I IFN, we also explored if these MP take up *P. yoelii* (*Py*)-infected RBCs, and provide STING-dependent signals to the pDC for full activation. Conclusions of this work are that:

i) CD169^+^ MP uptake *Py*-infected red blood cells (iRBCs) by FACS and IVM.

ii) A majority of CD169^+^ MP in the BM undergo activation and loss (>70%) between 30 and 36 hrs post infection, consistent with their implication in iRBC sensing and uptake (Revised Figure 1)

iii) CD169^+^ MP control pDC arrest. In *Py*-infected mice in which CD169^+^ MP are depleted, pDC fail to arrest (new Figure 2).

iv) TLR7 signaling in pDC is also required for their arrest and augmented interactions with the MP (revised Figure 3). v)Homing of pDC to CD169^+^ MP and acquisition of type I IFN producing capacity requires G_I_-mediated chemotaxis (revised Figure 4).

v) STING signaling in CD169^+^ MPs is required to enable pDC production of type I IFN and for their egress from the BM (revised Figure 5 and new Figure 6).

The revised version streamlines the original message and includes many new experiments and data that directly support the conclusions outlined above. We have added a new set of experiments that directly link CD169^+^ MPs to pDC arrest during *Py* infection (new Figure 2). We have generated novel partial bone marrow (BM) chimera mice to demonstrate that (i) TLR7intrinsic signaling in pDC is required for pDC arrest (revised Figure 3) and (ii) pDC-extrinsic STING-signaling regulates their retention in the BM (revised Figure 5). We also produced novel conditional knockout mice in which CD169^+^ MP lack STING. With these mice we established that STING signaling intrinsic to CD169^+^ MP is required to prime pDC production of type I IFN and to allow pDC egress from the BM (new Figure 6).

2. From Figure 1, the authors conclude that CD169 is involved in the binding and uptake of infected RBC. However, the authors mis-state what the data show. They state "we found that CD169 het mice, BM MPs had significantly higher frequencies of both extra and intracellular TAMRA-Red+…" However, Figure 1D shows a lower frequency of extracellular iRBCs, but no difference in intracellular RBCs. Though the "representative" flow is picked to show a >5x difference in extracellular RBCs, the number of both intra and extracellular RBCs bound/internalized by macrophages is significantly different, showing that the number of BM MPs is what is changing. In fact, the authors show the F4/80hi cells go down in number after infection in Figure 2, consistent with this interpretation.

We agree that the data presented in Figure 1D was confusing. The labelled uninfected and infected RBC uptake experiments followed a different set up than that of the natural infection which was depicted in Figure 1A (now in revised Figure S1B). Briefly, RBCs (~5x10^7^) were collected either from uninfected mice or from *Py*-infected mice when they reached ~20% blood parasitemia (~3-4 days post infection). RBCs were next labelled with the TAMRA-red and transferred to naïve mice. Six hrs later, we (i) harvested the BM cells and quantified TAMRAred^+^ RBCs associated with the BM MP populations by FACS and (ii) conducted IVM on the BM of recipient living mice. With such experimental setting we focused on the RBC uptake phase before MP loss occurs (revised Figure 1C). The data show (by FACS and IVM) that CD169^+^ MP (both CD169^hi^ and F4/80^hi^) are involved in the uptake of *Py*-infected but not uninfected RBC (revised Figure 1A, B and Movie S1). This has also been clarified in the revised manuscript.

CD169 is a sialoadhesin known to bind RBCs. While our original data supported the hypothesis that CD169 is involved in *Py*-iRBC uptake, we decided to remove these data from the revised version in an effort to streamline the manuscript and focus on the most important take home message.

3. Much of the manuscript uses CD169-/- mice and the authors state that the CD169hi macrophages still exist in these mice, however it is critical to show this to be able to interpret the data using these mice as the authors have already shown that these cells are critical for pDC clustering and IFNa production.

The experiments using the CD169^-/-^ mice (prior Figures 1D and 2D) have been removed from the revised paper, following the rationale outlined above. The gating strategy we used throughout the manuscript to identify BM CD169^+^ MPs is depicted in revised Figure S1A.

4. In Figure 2, there are two contradictory pieces of data. In Figure 2A, CD11a is show to be downregulated in infected mice, while in Figure 2D, CD11a is unchanged compared to uninfected mice.

We have pooled 2 independent replicate experiments now presented in the revised Figure S1D.

Overall, we found a consistent downregulation of CD11a expression on CD169^+^ MP (CD169^hi^ and F4/80^hi^ MP subsets). Additional replicate experiments showed the same result, but could not be pooled as a result of staining (MFI) differences, so we are confident these data are representative.

5. Although the authors acknowledge that PT treatment has indirect effects, they still conclude that pDC motility is required for pDC IFNa production and fail to address the possibility that motility of other cell types, including CD169+ macrophages, being involved.

We certainly agree with this reviewer that whole body PT treatment is not ideal and may affect other cells’ motility. One possibility is to treat purified pDC with PT in vitro and transfer them to naïve or infected hosts. However, adoptive transfers of pDC is very ineffective (they only represent less than 0.5% of DC) and we could not find transferred pDC in the BM (a point also noted by Reviewer #3). Another approach would involve expression of PT in pDC, but we are not aware of a pDC-specific inducible promoter we could use for this purpose, therefore this seemed beyond the scope of the current work. We also note that CD169^+^ MP were not motile in our IVM experiments. This observation has been quantified and is now shown in revised Figure S2B. In light of these comments, we have nevertheless carefully re-worded this section to reflect such limitations.

6. In Figure 7, the authors demonstrate that STING is required independently of cGAS for efficient type I interferon production by pDC. However, the authors had already demonstrated a role for STING extrinsic to pDC in their previous publication. Therefore, the only new piece of data here is that it is independent of cGAS, which is interesting, but needs to be followed up on in its own right.

In our previous work, we had established that STING signaling extrinsic to pDC and in the radiosensitive compartment, was important for pDC to secrete type I IFN. In the revised paper, we generated novel conditional knockout mice in which CD169^+^ MP lacked STING (new Figure 6). The data show that STING signaling in CD169^+^ MP is required for pDC production of type I IFN and egress from the BM. We further confirmed that when STING lacked only in the radioresistant compartment, pDC formed large clusters and accumulated in the BM (revised Figure 5H).

We have removed the cGAS data from the revised paper as we agree that this would need further investigations, and currently distracts from the main message of this work.

7. This manuscript would be greatly enhanced by a model of what had been shown previously and what is shown in this manuscript.

This is a great suggestion, and we provide a schematic to further clarify this aspect below. We have incorporated the final model in Figure 6D of the revised paper.

8. Use of statistics throughout the manuscript is not consistent and tests used are not noted in figure legends. Sometimes only 2 groups in a figure are compared and sometimes all groups, even within one figure panel, with no justification.

We apologize for this, and have now carefully checked that all statistical analyses were done and that the tests used were specified.

Reviewer #2:Moore et al. characterize the phenotype and movement of pDCs in the bone marrow of infected mice, and test several pathways (TLR7, STING, cGAS, GPCRs) for their role in regulating pDC motility, phenotype, and interferon production. There are several nice experiments here. Unfortunately, the authors grossly misrepresent or overinterpret their findings and extrapolate to a model of cell-cell interaction for which they provide little evidence. This is clearly a follow-up study to their 2016 PLoS Pathogens paper, but I'm not convinced it stands on its own; at a minimum it needs extensive textual revision to accurately represent the data, and once it has done that, I'm not sure it meets broad enough interest for an eLife paper.

We thank this reviewer for her/his time and critiques. Following up on these comments and those from the other reviewers, we have conducted many additional experiments to address key weaknesses. We have also significantly revised the manuscript to incorporate these new experiments (two new figures and four substantially revised figures), and clarify what is new in this work compared to our previous report (please also check our response to Reviewer#1 comment #1 on this major point, and the new schematic provided above and in revised Figure 6D). Detailed answers to the specific critiques of this reviewer are further provided below.

My main concern is that the text misrepresents what the data actually show. E.g., the title and abstract claim that CD169+ macrophages coordinate the localization and function (IFN secretion) of pDCs in a STING- and TLR7-dependent manner, but there is not a single figure in the paper that quantifies interactions between macs and pDCs. (The handful of microscopy images are not sufficient to make conclusions about these interactions-and the authors do not even attempt to do this, e.g. by using arrowheads to mark interacting cell clusters). The authors do state (p. 9) that pDC did not cluster around CD169+ macs in infected TLR7 KO mice compared to WT mice, but they show no data, and cite their previous paper for the WT observation.

We have added new/revised figures in which:

i) pDC dynamic behavior in the presence or absence of CD169^+^ MPs has been quantified in naïve versus *Py*-infected mice (new Figure 2A-D). We now also link the presence of CD169^+^ MP to pDC arrest during *Py* infection. When CD169^+^ MP are depleted, pDC fail to arrest and cluster.

ii) In *Py*-infected TLR7^-/-^ mice, pDC slow down but never arrest like in WT mice, therefore they do not form clusters. The new sets of experiments assess how STING and TLR7 signaling in pDC affect their motility (revised Figures 3, 5 and our answer below). We have also developed new scripts to quantify pDC and MP interaction by measuring their contact times (revised Figures 3G, 5F).

Further, there are no data to show that STING and TLR7 are acting within macs, rather than pDC (or another cell type), to affect arrest or clustering of pDC. I gather that the authors may have shown in their 2016 paper a requirement for mac-expressed STING in pDC production of interferon, but that doesn't mean the same is true for pDC motility/ clustering. (And it doesn't tell us anything about the TLR7 requirement.)

The revised Figures 3 and 5 incorporate new IVM experiments using partial BM chimera models in which we could visualize pDC lacking TLR7 or STING in WT host mice. When pDC lacked TLR7, they failed to arrest and interact with CD169^+^ MP (Figure 3E-G). In contrast, when pDC lacked STING, they arrested and interacted with the MP like WT pDC (Figure 5E, F). These results therefore (i) confirm our original observations in TLR7^-/-^ and STING^-/-^ mice, and (ii) directly link pDC motility (arrest, clustering) and interaction with CD169^+^ MP to their ability to secrete type I IFN (shown in our prior study).

The authors establish in Figure 1 that macs take up infected RBCs, but they do not show that pDCs do NOT take up iRBCs (which could perhaps demonstrate indirectly that pDCs must get their activating signals from macs). There is precedent for direct pDC uptake of malaria parasites (Wykes et al. PNAS 2011), though controversial… still, in this paper at least, the authors have not demonstrated that any interactions with CD169+ macs are controlled by STING/ TLR7/GPCR, or required for IFN production by pDCs.

We found no evidence for pDC uptaking *Py*-iRBCs neither in our FACS uptake experiments (See revised Figure S1C), nor in IVM. The new experiments shown in the revised manuscript establish that CD169^+^ MP and TLR7 sensing in pDC are both required for pDC arrest and clustering. We also show that STING sensing in pDC do not play a role in their arrest, clustering or production of type I IFN. Furthermore, using a new set of conditional knockout mice we generated, in which CD169^+^ MPs lack STING, we report that pDC fail to secrete type I IFN and egress from the BM during the infection (new Figure 7), as observed in whole body STINGdeficient mice (revised Figure 5I). Together with the new data presented in the revised Figure 2E-G, where we link the presence of CD169^+^ MP to pDC arrest, the results support the original model that CD169^+^ MP provide STING-dependent signal(s) to prime pDC production of type I IFN and enable their egress from the BM.

I also think the involvement of TLR7 in pDC arrest-a key point of the paper-is overstated. The section heading (p.8) states that "pDC arrest in Py-infected bone marrow is TLR7-dependent." But as the authors note, TLR7 KO pDCs have much higher baseline activity. They measure motility in four ways: track velocity, displacement rate, MSD, and slope of MSD. The latter two clearly show that TLR7 KO pDC still arrest after infection; the stats are not there to show whether the magnitude of their arrest is significantly different from WT. One measurement of motility (velocity) shows that TLR7 KO cells do differ from WT in that they don't slow down after infection. The final measure, displacement rate (3C- bottom) is not convincing: WT mice decrease displacement rate significantly after infection and TLR7 KO mice don't, but the WT infected group has an n of 7, and the TLR7 group only has 3 mice and is trending downward. It seems very possible that with more mice, this would become significant. So. at least two, and possibly 3 of the 4 measures of motility show NO difference between WT and TLR7-/- pDC. Given this, I think the finding is way overstated. The authors can still say that TLR7 plays some role in pDC motility, but they need to qualify this quite a bit (which they do on p. 9, but not throughout much of the paper, including the abstract).

This point is well taken. We have conducted several new experiments to address these weaknesses (revised Figure 3). In addition to analyzing pDC motility in additional TLR7^-/-^ mice, we have also generated partial BM chimeras in which only TLR7^-/-^ pDC can be visualized in a WT host environment, allowing to assess if TLR7 signaling intrinsic to pDC alters their motility. While in uninfected mice, TLR7^-/-^ pDC exhibited higher motility than in WT counterparts (as measured by TV, DR, MSD, revised Figure 3), the most important finding was in *Py*-infected mice where TLR7^-/-^ pDC, while slowing down, remained as motile as in uninfected WT mice. This was in contrast to WT pDC that arrested compared to that of uninfected mice. We have added statistical analyses comparing all groups for each parameter measured. Moreover, by quantifying the contact time between pDC and CD169^+^ MP, we show an increase only for WT but not TLR7^-/-^ pDC, consistent with this finding. This new set of experiments therefore strongly support the idea that TLR7 signaling in pDC control their arrest and activation.

A third unsupported conclusion is that "PT treatment led to immobilization of pDC, and their lack of visible clustering with MP" (p. 10, Figure 5A, B). The authors show that pDCs are largely nonmotile in infected, PT-treated mice, but they do NOT compare to untreated controls, in which we know that pDCs are arresting and largely nonmotile (Figure 3A second panel). Is there a difference between motility in infected, PT-treated versus untreated mice? This is critical to show.

We thank this reviewer for this suggestion. We have quantified pDC motility in PT-treated or untreated *Py*-infected mice (revised Figure 4A, B and Supplementary movie 7). The data show that DR of pDC in PT-treated mice was significantly reduced compared to that of untreated, showing that blocking G_I_-protein signaling and chemotaxis, induced stronger immobilization of pDC than *Py* infection only.

Further, as mentioned before, there is no measurement of clustering with macs, so the authors should not assert that PT treatment alters this.

The revised Figures 3G and 5F provide a quantification of pDC and CD169^+^ MP contact times.

However, in PT-treated mice, pDC arrest was drastic compared to untreated mice (revised Figure 4A, B and new Supplementary movie 7), making it unlikely that pDC and MP can establish functional interactions. We have carefully re-written this section of the results to highlight these findings.

Even the assertion that pDC are "morphologically distinct" after PT treatment seems overstated (p. 10); Figure 5C shows no difference in shape factor and a trending but non-significant difference in longest axis.

We have conducted further analyses of these data using software package Volocity 6.3 to include the monitoring of more pDC (revised Figure 4D). The data reveal statistically significant differences in pDC from PT-treated and untreated, *Py*-infected mice, using multiple metrics related to pDC shape including Longest Axis, Surface Area and Shape Factor. The Shape factor is a numerical indication of how similar 3D shape is to a perfect sphere with shape factor 1 indicating a perfect sphere.

Reviewer #3:pDC are important cells in the innate immune system that produce type I IFN (IFNab) in response to infection or injury. This group published in 2016 that pDC are highly motile in the bone marrow, but arrest in proximity to macrophages in the context of experimental malaria. Here, they further investigate the role of the CD169 receptor, TLR7, STING and cGAS innate sensors and GPCR motility receptors on this response. The work more clearly defines the role of the innate sensing pathways and chemokine receptors in the motility phenotype and the functional endpoint of IFNab production. There are a number of very nice aspects to this study. They have developed a robust method to detect the selective interaction of parasitized RBC with macrophages and to assess whether or not they are internalized on captures on the surface of macrophages. The baseline motility is completely pertussis toxin sensitive, although it was not possible to link the deceleration to an IFNab responsive chemokine receptor/chemokine pair. The interesting result is that the control of motility is necessary for the response, but not sufficient for IFNab production. A weakness of the paper is that the genetic manipulations effect both the macrophages and the pDC, so it's not possible to determine, for example, if TLR7 or STING are mainly operating in the pDC or the macrophage, or both.

We thank this reviewer for his positive comments and fair assessment of our work. The revised manuscript now includes new experiments in which we show that CD169^+^ MP control pDC arrest (new Figure 2). We also generated partial BM chimeras to visualize TLR7^-/-^ or STING^-/-^ pDC in a WT environment (revised Figures 3, 5). Lastly, we have produced new conditional knockout mice in which STING is lacking in CD169^+^ MPs (new Figure 6). These new data confirm the original phenotypes and establish that:

i) CD169^+^ MP are needed for pDC to arrest during *Py* infection (new Figure 2).

ii) TLR7 sensing and signaling in pDC is also required for pDC to arrest, cluster and interact with CD169^+^ MP (revised Figure 3).

iii) STING signaling in pDC is not important for them to arrest (revised Figure 5).

iv) STING signaling is important in CD169^+^ MPs to provide STING-dependent signal(s) to pDCs so that they can initiate type I IFN secretion. The data also show that if such signal(s) is(are) missing, pDC accumulate in the BM, form larger clusters and fail to egress from the BM (revised Figures 5 and 6).

The number of experiments is not quite enough to nail down the pDC migration phenotype of the TLR7 deficient mice. Three of the 4 mice analysed have more persistent migration such that the MSD is clearly greater. While the trend is clear the statistical significance is marginal, but this seems to be due to a limited samples size. So a couple of more experiments would be helpful. The data on the STING KO confirm that they can nail this when the migration phenotype is similar between two mouse strains.

The revised paper includes additional IVM experiments to strengthen the original phenotype, as well as all the new partial BM chimera experiments that confirm and extend our original findings and cell-intrinsic phenotypes.

[Editors’ note: what follows is the authors’ response to the second round of review.]

Reviewer #1:This is a revised manuscript by a group that has published on this topic before in 2016 (Plos Pathogen paper). They largely extend their previous findings and add some mechanistic studies demonstrating the requirement of CD169+ macrophage intrinsic STING signaling in regulating pDC motility/arrest. Furthermore their data suggest that adequate type I IFN production by pDC may also require macrophage intrinsic STING signaling.Since I was not part of the initial review I evaluated the previous reviewer comments and assessed how well the authors responded to the criticisms. I agreed with the majority of the concerns raised by the previous reviewers. Furthermore, the authors have attempted to strengthen the study in good faith and have included new studies. In particular, in inclusion of experiments where CD169 conditional STING-KO mice are used adds to the novelty of this study.The study is largely an extension of their previous work and in the revised manuscript the results for the most part support their conclusions. The use of dynamic imaging to assess pDC dynamics in the bone marrow is interesting and will add new avenues for further studies.This study also adds to our knowledge of how pDCs get activated and produce type I IFNs in vivo. The mechanisms of how pDCs get activated in vivo still remain poorly understood, however, this report begins to shed some light.

We thank this reviewer for his time and thorough assessment of our revised work.

Some deficiencies still remain, which will be discussed below:Concerns that still remain:1. Can the authors comment on the concern that use of anti-CD169 antibody for imaging the macrophages in vivo may prevent adequate uptake of infected RBCs or parasites in vivo?

We have conducted a new experiment where we quantified the uptake of labelled iRBC in mice injected with anti-CD169 or control isotype mAbs (new Figure 1—figure supplement 2B). Data confirmed the greater uptake in F4/80^hi^ MP compared to CD169^hi^ MP, but did not show any differences between anti-CD169 or isotype mAb-treated mice, suggesting that labelling CD169^+^ MP with the anti-CD169 mAb in vivo does not interfere with MP uptake of *Py*-iRBC.

2. Could the authors comment more on the physiological significance of type I IFN during malaria infection. Are pDCs needed for clearance of the infection and if type I IFN is required at all for the benefit of the host. This is a rather controversial topic and a clearer discussion taking their own studies in to account is warranted.

Type I IFN enhances inflammation and lethal outcomes in the *P. yoelii YM* (Spaulding et al., PLoS Pathogens 2016) and in the *P. berghei* (Sharma et al., Immunity 2011) models of lethal infections. It also prevents effective control of non-lethal *P. yoelii* (17XNL) and *P. chabaudi chabaudi AS* by preventing optimal Tfh and GB B cell responses as a result of inadequate inflammation (Zander et al., PLoS Pathogens 2016 ; Sebina et al., PLoS Pathogens 2016). Overall, these studies are consistent with the idea that type I IFN are detrimental in malaria infections. While in our study, we reported that pDC were the major producers of type I IFN during *Py YM* infection, pDC depletion only mildly rescued mice survival (Figure S3E in our PLoS Pathogens 2016), suggesting type I IFN from other cells significantly contributed to the lethal outcome. However, and as pointed out by this reviewer and in our original discussion, results from Yu et al. (Immunity 2016) differed from ours and from the general consensus. These authors reported that STING deficiency in pDC led to higher secretion of type I IFN as a result of the loss of SOCS1-mediated inhibition of TLR7/MyD88 signaling, and to greater resistance to malaria infection. They also revealed that pDC were required for type I IFNdependent mouse survival, a finding consistent with their model but discrepant with the broader consensus. We have further expanded and clarified this section of the discussion accordingly (page 20).

3. The fact that the motility of pDCs in TLR7 KO mice even under a steady state is so much higher still makes the interpretation of the data in Figure 3 difficult.

Indeed, it is notable that pDC are hyper motile in TLR7-deficient mice, suggesting that tonic TLR7 signaling may be regulating pDC motility rates at steady state. While infection does lower motility of TLR7-deficient pDC in a cell autonomous fashion, they never arrested like WT pDC, by all 3 measures (TV, DR and MSD). This suggested that other factors are also controlling pDC speed, but that these are insufficient to completely arrest these cells. As our previous work found that TLR7-deficient pDC cannot produce type I, slowing these cells during infection is insufficient for proper activation. Taken together, these results suggest that reduced motility is insufficient to lead to pDC activation and only complete arrest, with TLR7 signaling, can lead to proper activation. As another way to convey this finding, we have also represented the data by compiling the proportion of WT versus TLR7^-/-^ pDC arrested or moving in the various experimental settings, using TV and DR threshold from WT mice (new Figure 3-supplement figure 1A). The data show a comparable proportion of pDC moving or arrested in *Py*-infected *Tlr7^-/y^* mice and naïve WT or *Tlr7^-/y^* mice, in contrast to *Py*-infected WT counterpart.

4. The authors make the claim that CD169+ macrophage intrinsic STING signaling is required for the production of Type I IFN by pDCs. However, they only use the reporter mouse for these studies, and the imaging is not very clear. Thus, the authors should show ELISA data in the serum and bone marrow to complement their imaging studies.

In our 2016 study, we established that the *IFNb^YFP^* reporter mouse faithfully tracks for type I IFN production by pDC, both for IFN and IFN in the serum and BM, as quantified by ELISA. We have clarified this aspect in the revised manuscript (pages 14-15).

5. The CD169+ macrophages themselves can produce copious amounts of Type I IFNs. Can the authors eliminate the possibility of paracrine IFN secreted by the macrophages as another mechanism by which pDCs respond to and produce more IFN – especially in studies where CD169-DTR mice are used or in STING conditional KO studies? Here too, assessing IFN production will be valuable.

It is very much a possibility that we considered, and further emphasized in the revised discussion (page 19). STING leads to TBK/IRF3 activation and IFNβ transcription, thus IFNβ produced by CD169^+^ MP via STING, could enhance pDC production of type I IFN. Depleting CD169^+^ MP prevents pDC to produce type I IFN (our 2016 study, using the reporter and ELISA) and we do not detect *Ifnb* gene transcription in CD169^+^ MP. In addition, removing type I IFN receptor on pDC is known to prevent them from secreting more type I IFN (Asselin-Paturel et al., JEM 2005). Also, we do not know of conditional *Irf3 Irf7* double knockout mice that we could cross to *Cd169^Cre^* expressing mice to abrogate any type I IFN production by CD169^+^ MP. Thus, we feel that these important limitations in the experimental systems readily available to us, make the formal demonstration of the STING-dependent mechanism by which MP activate pDC largely beyond the scope of the current study.

6. Finally, the authors should be careful about making any substantive conclusions with respect to pDC – CD169+ macrophage interactions. In this respect the dynamic imaging is not as convincing. The authors themselves admit that due to the high density of macrophages in the BM "it makes it difficult to establish a functional interaction"

Our prior work showed that deletion of these MP prevented the production of type I IFN by pDC. The first revision of this manuscript incorporated new sets of experiments in which we directly investigated whether CD169^+^ MP regulated pDC motility (Figure 2E-G). These new data clearly established that selective depletion of CD169^+^ MP prevented pDC arrest during infection, directly linking the presence of CD169^+^ MP to pDC stopping. While in vivo imaging can never fully rule out that these functional interactions are indirect, or that an additional component is involved, it is more likely than not that they are linked, until new information supports an alternative model. We have now clarified these notions and revised the text to avoid any overstatements.

Reviewer #2:This revised version investigates the role of CD169 BM macrophages in the activation of pDC following the injection of P.yoelii infected RBCs. It aims at demonstrating that BM CD169+ macrophages are key regulators of pDC retention and activation, in a TLR7 and STING dependant manner.Overall, the take home message appears valid but some experiments need to be performed in order to exclude technical artifacts:

We thank this reviewer for his time and evaluation of our work.

1. TAMRA and PE appear to have very similar excitation and emission spectra. In addition, they both emit in the "Red" channel, which is typically highly autofluorescent. Why did the authors chose these two dyes to specifically label RBC and CD169+ Macs? The use of green/red (or blue/red or blue/green) fluorochromes might have improved the quality of the movies.

We apologize for the confusion between the text and the figure legend. In the IVM uptake experiments presented in Figure 1B and Figure 1-video 1, *Py*-iRBC were labeled with TAMRA-Red while CD169^+^ MP were labeled with anti-CD169-FITC mAb, allowing for easy discrimination between CD169^+^ MP (green) and *Py*-iRBC (red). In the FACS uptake experiments presented in Figure 1A, transferred *Py*-iRBC were also labeled with TAMRA-Red and CD169^+^ MP were stained ex vivo with anti-CD169-APC and F4/80-FITC*.* In all other IVM experiments of the paper, pDCs constitutively expressed GFP^+^ (PTCRA-eGFP) and CD169^+^ MP were labelled in vivo using anti-CD169-PE mAb injection. We have further clarified these technical details in the revised manuscript (revised text, Figure 1B and Figure 1—figure supplement 1B legends, and in the Materials and methods).

2. Reviewer 1 is correct when he/she states that PT treatment has indirect effects than can alter pDC motility and IFNa production. As the authors fail to identify the chemokines (or PT sensitive signalling pathways) that are at play, it is incorrect to state that pDC motility is required for pDC IFNa production. The authors should make this very clear in the manuscript.

We have carefully edited the revised version to reflect such limitations.

3. Figure 1 does not convince me that CD169 Macs pick up infected RBC. It is well documented that CD169+ macs are fragile (at least in lymphoid organs) and that most CD169+ cells analyzed by flow are not even macrophages (PMID: 22675532 and more recently PMID: 34818538). As IVM lacks resolution, the authors should perform the same experiment in a PTCRA-eGFP mouse and stain fixed bone sections for CD169/GFP expression and then quantify the amount of fluorescent RBC captured by pDC and CD169+ macs. In addition, as the authors possess CD169Cre mice, I am wondering why they did not cross this strain to a fluorescent reporter such as the Rosa-RFP line to (a) visualize CD169+ macs by IVM and (b) prove that pDC are not fate-mapped in this model, and thus not targeted in CD169DTR mice.

We have crossed the *Cd169^Cre^* mouse to both *Rosa26^LSLYFP^* and *Rosa26^LSLTd^* mice, as requested by this reviewer (new Figure 1—figure supplement 1C). The data compare both reporters and the in vivo anti-CD169 mAb labelling of CD169^+^ MP. In all three settings, we observe cells expressing low and high levels of CD169. Only the high expressors corresponded to CD169^+^ MP (>80%) while the low expressors likely resulted from non-specific staining or expression (YFP or tomato) of CD11b^+^ cells. This suggested that in vivo anti-CD169 mAb labelling is as effective to track CD169^+^ MP as the reporter systems are. Thus, we do not believe that redoing all IVM experiments in *Cd169^Cre^* reporter mice would change any of the major findings.

As far as pDC targeting in the *Cd169^DTR^* mouse, we have reported in our 2016 study (Figure S6A) that DTtreatment in uninfected and in *Py*-infected *Cd169^DTR^* mice did not decrease pDC proportion in BM, blood and spleen. The finding that pDC proportion and numbers (current Figure 7B) are significantly increased in the BM of DT-treated *Py*infected *Cd169^DTR^* mice is consistent with our the results showing increased retention of pDC in the BM of STINGdeficient mice (Figure 5I) and in mice where CD169^+^ MP lack STING (Figure 7B). Altogether, these results strongly argue against pDC targeting in the *Cd169^Cre^* or *Cd169^DTR^* models.

4. Related to the previous points, I am concerned by FigS2C and sup movie 3. Both are showing data obtained in CD169 DTR mice treated with DT. While flow cytometry showed a 6 fold reduction of CD169+ macs upon DT injection, IVM does not show a similar decrease. If true, it suggests that one technique is not reliable to observe CD169+ macs. Can the authors comment on this?

Thank you for noticing this. We went back to the original movie settings and realized that the two movies had distinct gain settings to favor the visualization of pDC. Unfortunately, postprocessing oversaturated signal in the PE (red) channel, making CD169^-^ autofluorescent cells appear PE^+^ in the DT-treated mice. We have reduced the gain in the re-processed corresponding movie without altering the gain (“version 2”) so that we can provide side by side with the prior version, together with the two different export settings. We also provide for this reviewer (Author response image 1) an example image projection of the expected anti-CD169-PE punctated staining (red arrows) in control versus DT-treated *Py-*infected mice, which is very different from that of red-autofluorescent cells (orange arrows).

Gain settings optimized for visualization of MP:

o RDX2 settings

Mix: 104

Max: 805.67

Gamma: 1.28

o RDX3 settings

Mix 128.98

Max: 4095.00

Gamma: 1.34

**Author response image 1. sa2fig1:**